# Motor planning under uncertainty

**Laith Alhussein[1], Maurice A Smith[1,2]***

[1]John A. Paulson School of Engineering and Applied Sciences, Harvard University, Cambridge, United States; [2]Center for Brain Science, Harvard University, Cambridge, United States

**Abstract** Actions often require the selection of a specific goal amongst a range of possibilities, like when a softball player must precisely position her glove to field a fast-approaching ground ball. Previous studies have suggested that during goal uncertainty the brain prepares for all potential goals in parallel and averages the corresponding motor plans to command an intermediate movement that is progressively refined as additional information becomes available. Although intermediate movements are widely observed, they could instead reflect a neural decision about the single best action choice given the uncertainty present. Here we systematically dissociate these possibilities using novel experimental manipulations and find that when confronted with uncertainty, humans generate a motor plan that optimizes task performance rather than averaging potential motor plans. In addition to accurate predictions of population-averaged changes in motor output, a novel computational model based on this performance-optimization theory accounted for a majority of the variance in individual differences between participants. Our findings resolve a long-standing question about how the brain selects an action to execute during goal uncertainty, providing fundamental insight into motor planning in the nervous system.

## Introduction

We often plan actions before knowing the exact goal. For example, in baseball, a pitch can take as little as 400 ms to reach the batter. With a reaction time of at least 200 ms and a 150 ms swing duration, the batter must initiate the swing based on visual information from only the first 10–20% of the ball's flight, amidst considerable uncertainty surrounding the ultimate timing and positioning of the ball's arrival at the plate. Nonetheless, batters deftly make use of this uncertain information and often make good contact with the ball. But what are the mechanisms that underlie this ability?

To study how motor planning proceeds when goal information is ambiguous, controlled experiments have introduced uncertainty in several different ways: by delaying goal information (*Chapman et al., 2010*; *Ghez et al., 1997*), pairing potential goals with distractor stimuli (*Arai et al., 2004*; *Walker et al., 1997*), displaying noisy visual cues (*Hudson et al., 2007*; *Resulaj et al., 2009*), or incorporating high-level cognitive decision-making events during movement (*Song and Nakayama, 2009*). In response to this uncertainty, which prompts consideration of multiple potential goals, movements frequently begin in directions that are intermediate between them. These intermediate movements – widely considered a telltale sign of motor planning under uncertainty (*Chapman et al., 2010*; *Hudson et al., 2007*; *Stewart et al., 2014*; *Wong and Haith, 2017*; *McPeek and Keller, 2004*; *Tipper et al., 1998*; *Gallivan et al., 2011*; *Hening et al., 1988*; *Favilla et al., 1990*) – are thought to provide fundamental insight into the neural processes by which the brain prepares an action to achieve a desired goal (*Chapman et al., 2010*; *Arai et al., 2004*; *Stewart et al., 2014*; *Wong and Haith, 2017*; *McPeek and Keller, 2004*).

The prevailing interpretation of intermediate movements during uncertainty is that they represent an average of the individual motor plans that arise from each potential target (*Chapman et al., 2010*; *Ghez et al., 1997*; *Arai et al., 2004*; *Stewart et al., 2014*; *Tipper et al., 1998*; *Gallivan et al., 2011*; *Hening et al., 1988*; *Favilla et al., 1990*; *Gallivan et al., 2016*;

**\*For correspondence:**
mas@seas.harvard.edu

**Competing interests:** The authors declare that no competing interests exist.

*Gallivan et al., 2017*; *Gallivan and Chapman, 2014*; *Stewart et al., 2013*; *Chou et al., 1999*; *Van der Stigchel et al., 2006*; *Eggert et al., 2002*; *Findlay, 1982*; *Godijn and Theeuwes, 2002*; *Ottes et al., 1984*; *Sailer et al., 2002*; *Coren and Hoenig, 1972*). This idea, termed motor averaging (MA), has been reported to occur for a number of key movement-related variables including a movement's direction (*Chapman et al., 2010*; *Ghez et al., 1997*; *Van der Stigchel et al., 2006*), path shape (*Stewart et al., 2014*; *Gallivan and Chapman, 2014*), effector orientation (*Stewart et al., 2013*), and even the feedback gains modulating how the motor system responds to errors induced by noise and external perturbations (*Gallivan et al., 2016*). Moreover, MA has been reported in a diverse array of behavioral paradigms, including point-to-point reaching arm movements (*Chapman et al., 2010*; *Gallivan and Chapman, 2014*), saccadic eye movements (*Arai et al., 2004*; *Van der Stigchel et al., 2006*; *Coren and Hoenig, 1972*), and isometric force control (*Hening et al., 1988*). In line with the MA hypothesis, neurophysiological work has suggested sensorimotor activity reflecting the parallel planning of movements to different potential targets (*McPeek and Keller, 2004*; *Cisek and Kalaska, 2005*; *Cisek, 2007*; *Baumann et al., 2009*; *Coallier et al., 2015*; *Pastor-Bernier and Cisek, 2011*), with the assertion that parallel plans are automatically combined via MA to form the final action plan (*Chapman et al., 2010*; *Stewart et al., 2014*; *Gallivan et al., 2011*; *Stewart et al., 2013*; *Gallivan et al., 2015*).

An alternative explanation for the occurrence of intermediate movements is that they instead reflect a deliberate plan that seeks to optimize task performance (*Hudson et al., 2007*; *Haith et al., 2015a*). A previous attempt to decouple this performance optimization (PO) hypothesis from MA demonstrated that during goal uncertainty intermediate movements are abandoned when high-threshold movement speed criteria are imposed or when potential goals have a wide spatial separation (*Wong and Haith, 2017*). These findings show that behavior at odds with MA can occur, and thus indicate that MA is not obligatory, which is a key result. However, because these tasks incentivize direct movements towards a single potential target for success, they are likely to promote a high-level explicit choice between targets before movement onset rather than low-level motor planning under uncertainty. Thus, the mechanisms underlying the formation of intermediate movements – a hallmark of movement planning during uncertainty – remain unclear.

It has been difficult to dissociate between MA and PO because the motor plan that leads to task success often resembles an average of individual-target motor plans (*Hudson et al., 2007*). Indeed, close examination of studies supporting MA reveals that the findings were, in fact, consistent with PO. For example, *Chapman et al., 2010* examined motor planning during goal uncertainty by employing a task where participants were asked to make rapid reaching movements towards one of several potential target locations, with the final target cued only after movement onset (i.e., 'go-before-you-know'). Analogously, *Gallivan et al., 2016* studied motor planning at the level of feedback control policies in an analogous go-before-you-know task, but used targets of different widths in order to modulate the gain of feedback responses (*Knill et al., 2011*). In both cases, intermediate actions were elicited when the goal was uncertain. In the Chapman study, the resulting movement was directed at the midpoint between the potential target locations, and in the Gallivan study, the resulting feedback gain was sized at the midpoint between the gains associated with each potential target. Although the authors interpreted these behaviors as evidence for MA, in both cases the results can also be explained by PO: in the Chapman study, initial movements towards the spatial midpoint brings the hand closer to all potential targets, thus reducing the size of the subsequent movement required to reach the final target once identified, and in the Gallivan study, intermediate feedback gains balance the cost of the effort needed for movements executed with high-feedback gains, against the likelihood that this effort will be necessary.

In another example, *Stewart et al., 2014* used the go-before-you-know paradigm to create goal uncertainty, but in combination with obstacles that were positioned to alter motor planning to one of the potential targets. In trials where the goal was initially uncertain, the obstacle placements resulted in intermediate movements that were deflected in a manner consistent with MA, and this was taken as evidence for the MA hypothesis. However, the obstacles were configured in such a way that the observed deflections on goal-uncertain trials, which were interpreted as evidence for MA, also happened to improve the safety margin around the obstacle. Therefore, PO for the goal-uncertain trials, which would result in motor planning that improved safety margins to minimize the likelihood of obstacle collisions, would also readily predict the experimentally observed deflections, calling the evidence for MA into question.

The interpretational and methodological issues of the studies outlined above are emblematic of the pervasive confound between MA and PO in studies examining motor planning under uncertainty. In the current study, we resolve the debate between these hypotheses by designing two sets of experiments that each perturbed the individual-target motor plans, as in previous studies (*Stewart et al., 2014*; *Gallivan et al., 2016*; *Gallivan et al., 2017*; *Nashed et al., 2017*), but in a manner that led to distinct predictions for MA and PO during goal uncertainty in both experiments. In one, we employ obstacle-based perturbations of motor planning, like in *Stewart et al., 2014*, but using novel obstacles that break the congruence between the predictions for PO and MA. In another, we create a novel dynamic environment that induces adaptive responses that make completely opposite predictions for PO vs. MA. In both cases, we find clear evidence that, when faced with uncertainty, humans form a motor plan that optimizes task performance.

## Results

To decouple motor averaging (MA) from performance optimization (PO) in both experiments (Expt 1 and Expt 2), we created modifications of the popular go-before-you-know task (*Hudson et al., 2007*; *Stewart et al., 2014*; *Wong and Haith, 2017*; *Gallivan et al., 2011*; *Gallivan et al., 2016*; *Gallivan et al., 2017*; *Gallivan and Chapman, 2014*; *Stewart et al., 2013*), where uncertainty is introduced on some trials by presenting participants with multiple potential reach targets and disclosing the final goal location only after movement onset. We designed paradigms that altered motor plans in such a way that the MA and PO theories would make contrasting predictions when the reach goal was uncertain. In Expt 1, we accomplished this by pre-training a force-field (FF) environment that physically perturbed 1-target movements to left and right lateral target locations with one FF environment (FF$_{LATERAL}$), but perturbed movements to a center target location with the opposite FF environment (FF$_{CENTER}$). Consequently, when the left and right target locations were presented as potential targets under uncertainty, MA predicts that intermediate movements incorporate the learned adaptive response to FF$_{LATERAL}$. However, PO predicts that these intermediate movements should be planned so that they travel towards the midpoint of the potential targets in order to maximize the probability of final target acquisition. This would, in contrast to MA, predict that intermediate movements incorporate the learned adaptive response to FF$_{CENTER}$, appropriate for center-directed movements, allowing us to decisively dissociate PO from MA.

In Expt 2, we achieved a second dissociation between MA and PO by placing a virtual obstacle that induced deflections as it blocked direct movements to one target. Consequently, when that obstacle-obstructed target was used as a potential target under uncertainty, MA predicts that intermediate movements inherit a portion of this deflection, even if it would steer them *closer to* the obstacle. In contrast, PO would predict that intermediate movements be consistently steered *away from* the obstacle to maintain a sufficient safety margin around it, as optimizing for task success would act to reduce the chance of obstacle collision. We thus present two distinct approaches that powerfully dissociate the predictions of the MA and PO theories for motor planning under uncertainty.

### Adaptation to novel physical dynamics can elucidate the mechanisms for motor planning under uncertainty

In Expt 1 (n = 16), we employed a version of the 'go-before-you-know' task in which a combination of different FF environments was used to investigate movement planning under uncertainty. While gripping a robotic manipulandum that could apply forces to the hand (*Figure 1a*), participants initiated 20 cm cued-onset reaching movements towards either a single prespecified target (1-target trials; *Figure 1b*, left) or two potential targets (2-target trials; *Figure 1b*, right). On 1-target trials, the target, located at a left (+30°), right (−30°), or center (0°) eccentricity from the midline, was displayed for 1000 ms before an auditory go cue was delivered. On 2-target trials, the same pair of potential targets always appeared in the left and right target locations for 1000 ms before the auditory go cue, but one (randomly selected) was extinguished 3 cm after movement onset, leaving only the final reach target on-screen. Comparing the data from 1- and 2-target trials thus allowed us to infer how uncertainty about the final target location influences motor planning on 2-target trials.

We designed a novel physical environment composed of multiple FFs that would result in very different predictions from MA vs. PO for motor planning on 2-target trials. Specifically, we trained

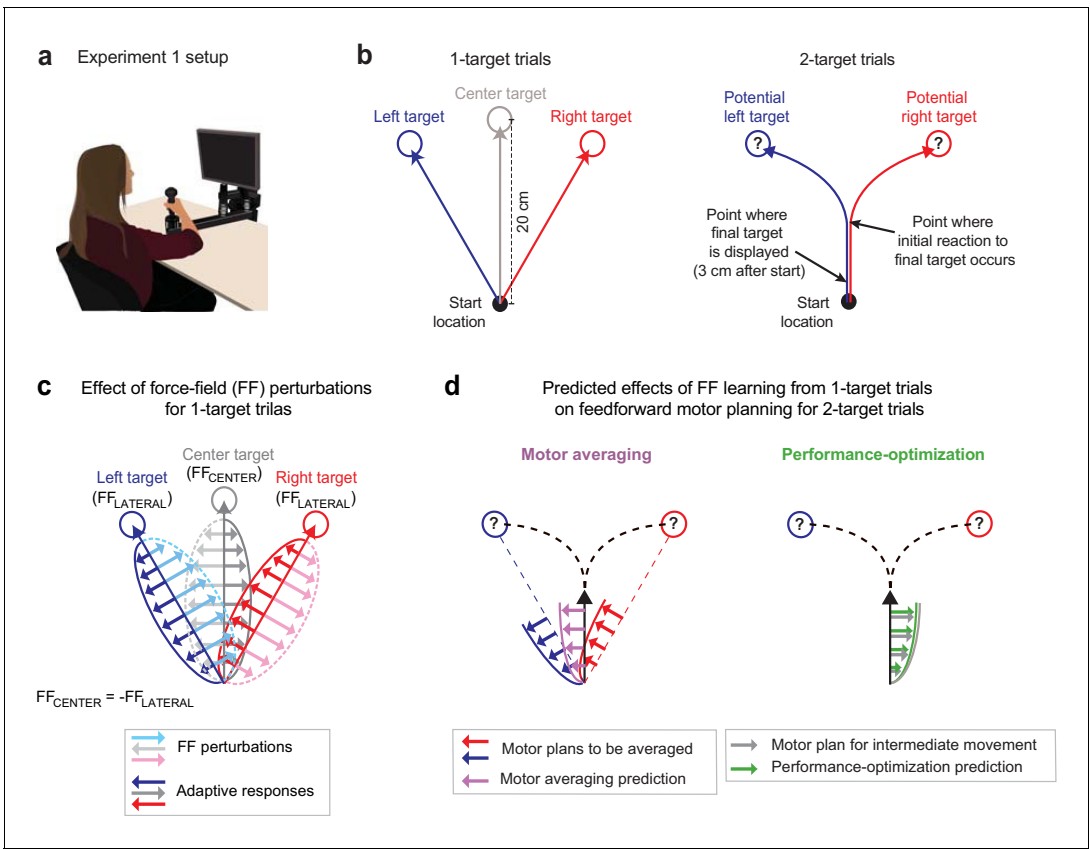

**Figure 1.** Multiple force-field (multi-FF) environment paradigm. (**a**) Setup for the multi-FF environment experiments. We altered environmental dynamics by exposing participants to viscous curl FFs in which the force vector perturbing reaching movements was proportional in magnitude and perpendicular in direction to the velocity of the hand (see Materials and methods). (**b**) Diagram of unperturbed 1- and 2-target trial types. On 1-target trials (left panel), a single target located in the left (+30°), right (−30°), or center (0°) direction was presented, and participants were instructed to initiate rapid 20 cm reaching movements to the target after an auditory go cue. On 2-target trials (right panel), a pair of potential targets, always in the left and right target directions, was presented before the go cue. Then, 3 cm after movement onset, we extinguished one target and highlighted the other to indicate the final goal. Delaying the precise goal information in this manner typically leads to initial reaching movements directed in-between potential target locations before participants produce in-flight movement corrections towards the final target. (**c**) Individual-target FF perturbations. During 1-target trials, we perturbed movements to the left and right targets with $FF_{LATERAL}$ (light blue and pink arrow and dotted trace sets, respectively) and the center target with $FF_{CENTER}$ (light gray arrow and dotted trace sets). The directions of $FF_{LATERAL}$ and $FF_{CENTER}$ were counterbalanced across participants, but always with $FF_{CENTER} = -FF_{LATERAL}$. Training in this composite environment alters the adaptive responses (darker solid arrow trace sets) in accordance with the FF imposed on each target. Note that the adaptive response on 1- and 2-target trials was measured as the force patterns participants produced on error clamp and partial error clamp trials, respectively (see Materials and methods). (**d**) Predictions for motor averaging (MA) and performance optimization (PO) for feedforward motor planning during uncertainty. Because both potential targets (left and right) were associated with $FF_{LATERAL}$, MA (purple arrows) predicts a force pattern consistent with $FF_{LATERAL}$ on 2-target trials. However, since the initial motion on these 2-target trials is in the direction of the center target, PO (green arrows) predicts the force pattern consistent with $FF_{CENTER}$, which is opposite the MA prediction.

participants to adapt to a viscous curl FF (see Materials and methods) that perturbed the left/center/right movements during 1-target trials with a $FF_{LATERAL}/FF_{CENTER}/FF_{LATERAL}$ composite environment (*Figure 1c*), where $FF_{CENTER} = -FF_{LATERAL}$, and the sign of the FFs was balanced across participants. For 2-target trials, MA would predict that participants average the force patterns ($FF_{LATERAL}$ in both cases) learned for the left and right (lateral) targets, which correspond to the potential target locations on 2-target trials (*Figure 1d*, left). In contrast, PO would predict that participants produce the force pattern ($FF_{CENTER}$) appropriate for optimizing the planned intermediate movement since this movement maximizes the probability of successful target acquisition (*Hudson et al., 2007*; *Haith et al., 2015a*; *Figure 1d*, right). Importantly, and in contrast to previous studies (*Gallivan et al., 2017*; *Nashed et al., 2017*; *Stewart et al., 2013*; *Stewart et al., 2014*), we were

able to reliably elicit intermediate movements during uncertainty in all our experiments by controlling the reward rate with sliding scales for movement time criteria (see Materials and methods).

Participants first performed baseline 1- and 2-target trials to gain familiarity with them, and then completed an FF training block, where we imposed the multi-FF environment on 1-target trials as outlined above. Note that, because we differentially perturbed movements to the center target compared to the lateral targets, adaptation to the FFs associated with the lateral targets, $FF_{LATERAL}$ in both bases, would interfere with adaptation to the FF associated with the center target, $FF_{CENTER}$. To elicit similar levels of FF adaptation for all three target locations, we included a greater number of training trials for the center target (using a ratio of 1:2:1 for left, center, and right target directions; see Materials and methods). The adaptive compensation to the FFs administered during training was measured with pseudorandomly interspersed error clamp trials (*Scheidt et al., 2000*). After training, participants experienced a test epoch, which included 2-target partial error clamp trials, on which the initial force pattern could be measured (see Materials and methods). The test epoch also included intermittent 1-target FF and error clamp trials, used to maintain the adaptive change in motor output developed during the training epoch and measure its ongoing state.

## Adaptive responses in a multi-FF environment are consistent with motor planning during uncertainty driven by performance optimization rather than motor averaging

Training within the multi-FF environment resulted in substantial levels of motor adaptation in all three target directions throughout both the training and test epochs as demonstrated in *Figure 2a*. We measured this adaptation with an adaptation coefficient, a regression-based metric for comparing each movement's force profile relative to the imposed FF perturbation (see Materials and methods) (*Smith et al., 2006*; *Sing et al., 2009*). We found that the final adaptation coefficient levels achieved at the end of the training period (defined as the last 10% of EC trials) were within ~9% for all three target directions (0.73 ± 0.15 [95% C.I.], 0.80 ± 0.15, and 0.73 ± 0.12 for training in the left, center, and right target directions, respectively). Importantly, these adaptation levels were largely maintained during the test epoch (0.71 ± 0.13, 0.68 ± 0.13, and 0.74 ± 0.09 for training in left, center, and right target directions, respectively). Therefore, we used the population-averaged force profiles measured on 1-target error clamp trials (i.e., the adaptive response) during this test epoch to construct our predictions for MA vs. PO, as shown in *Figure 2b*. Note that all force data are aligned to target cue onset ($T_{ON}$), the time at which one of the potential targets was extinguished for 2-target trials, which corresponds to the 3 cm excursion point shown in *Figure 1b*. Also note that all force profile data are shown for the time that feedforward motor output was observed on 2-target trials since forces due to feedback corrections do not reflect movement planning. We defined the time of feedback response onset ($T_{RESP}$) as the point in time where we could detect statistically significant feedback responses during 2-target trials (150 ms after target cue onset; see Materials and methods). In line with the adaptation coefficient measure we used to characterize overall learning, the more detailed analysis that we present of how force profiles evolve in time also normalizes raw force measurements by the level of the ideal compensation.

To specifically determine how movements are planned in uncertain conditions, we examined the force patterns predicted by MA and PO for 2-target trials and compared them to the force patterns we measured on 2-target trials (*Figure 2c*). The MA and PO predictions are essentially opposite, consistent with the $FF_{LATERAL}/FF_{CENTER}/FF_{LATERAL}$ environment we imposed during training. The MA prediction corresponds to the average of the force profiles (adaptive responses) associated with the left and right 1-target EC trials plotted in *Figure 2b*, and we note that because the left and right 1-target trial force profiles were remarkably similar, differential weighting for these motor plans would have little effect on the MA prediction. On the other hand, the PO prediction corresponds to the force profile associated with the center target plotted in *Figure 2b* as this is appropriate for optimizing a planned intermediate movement. Experimental data on 2-target trials show that participants systematically produce positive forces that are at odds with MA-based predictions, but in line with PO-based predictions for motor planning during uncertainty. We quantified the similarity between the 2-target trial data and the predictions using a prediction index that produces a value of +1 if the mean force from the experimental data were perfectly similar to the PO prediction, –1 if was perfectly similar to the MA prediction, and 0 if the data were halfway between both predictions (see Materials and methods). We measured this prediction index over two intervals. One extended out to

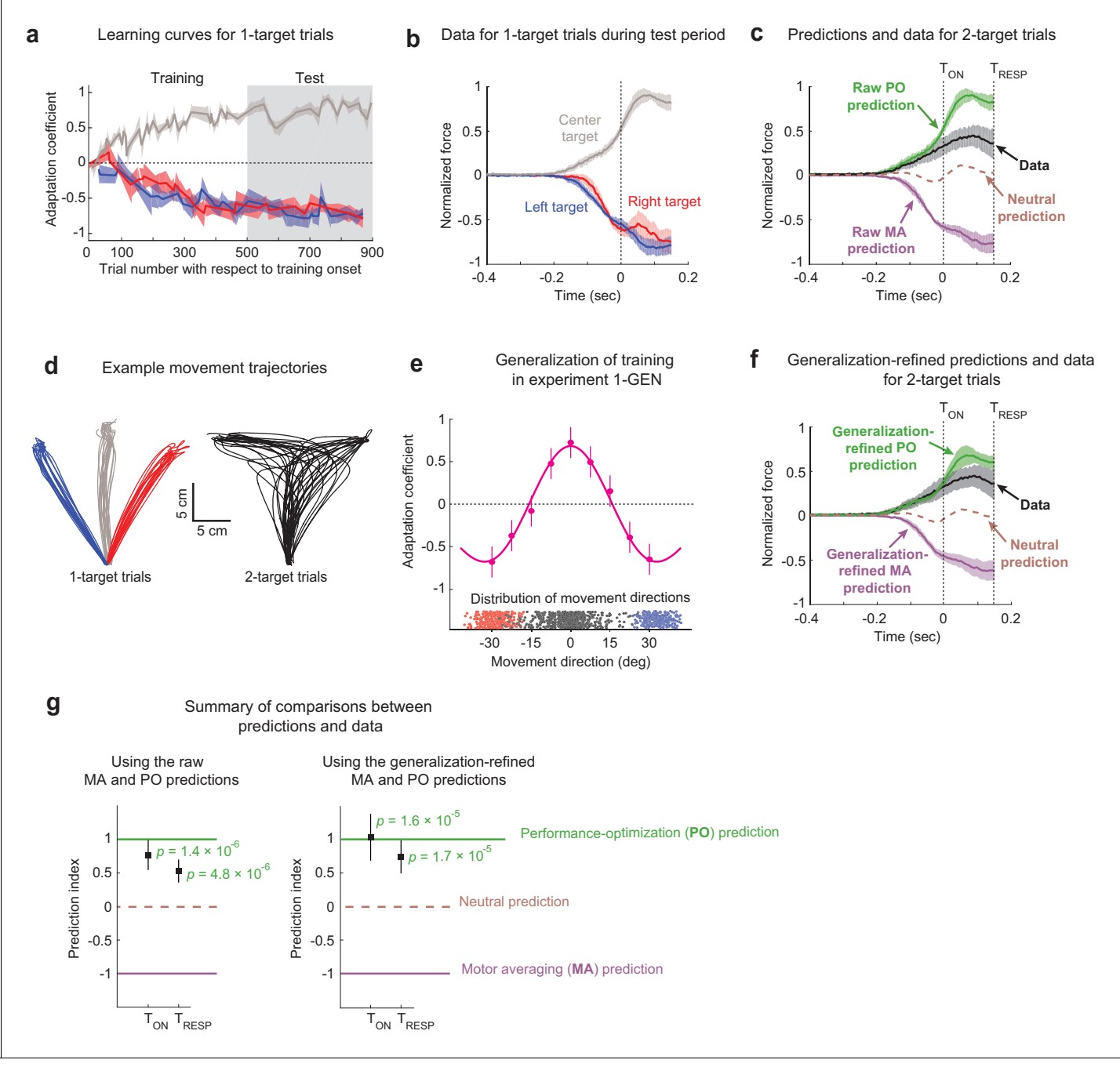

**Figure 2.** The effects of motor adaptation induced by novel environmental dynamics indicate performance optimization (PO) during uncertainty. (**a**) Learning curves for 1-target trials. Participants (n = 16) readily adapted left (blue), right (red), and center (gray) 1-target trial movements during the training period and maintained this adaptation during the test period. Note that experiments were balanced with +15/–15/+15 Ns/m force-fields (FFs) applied for the left/center/right targets in half the participants and −15/+15/−15 Ns/m in the other half, and that adaptation coefficients of +1 refer to ideal learning for the center target FF, and adaptation coefficients of −1 refer to ideal learning for the left/right target FFs. (**b**) Population-averaged force profiles from 1-target error clamp trials measured during the test period. Data is normalized so that +1 refers to the amplitude of the maximum force in the ideal adaptive response for the FF associated with the center target. (**c**) Raw predictions (green and purple) and force profile data (black) for 2-target partial error clamp trials. The raw PO prediction is based on the adaptive response for the center target, and the raw motor averaging (MA) prediction is based on averaging the adaptive responses for the two (right and left) potential targets. The midpoint between the MA and PO predictions provides a neutral prediction reference (brown dashed trace), indicating that the experimental data are more closely aligned with PO than MA. (**d**) Randomly selected trajectories from an example participant demonstrate non-trivial variability for movement directions in both 1- and 2-target trials. (**e**) Generalization data (magenta) from Expt 1-GEN (n = 10) is well characterized by a function obtained by summing Gaussians centered on each

*Figure 2 continued on next page*

*Figure 2 continued*

trained direction (+30°/0°/−30°). Blue and red dots indicate movement direction distributions for left and right 1-target trials, and black dots indicate the corresponding distribution for 2-target trials across all participants. (f) Generalization-refined predictions and force profile data from 2-target partial error clamp trials, where we used data from Expt 1-GEN to account for the effect of movement direction variability on the MA and PO predictions from Expt 1 (see Materials and methods). (g) Summary comparison between the data (black squares) and predictions (dashed lines). Both panels show a prediction index where −1 corresponds to the MA prediction and +1 corresponds to the PO prediction. This index was calculated by comparing the mean value for the force profile in the data to that from both model predictions, with means computed from the time of the go cue until either the target cue onset ($T_{ON}$) or when target-specific responses were first statistically detectable ($T_{RESP}$). The left panel shows results based on the raw predictions in panel (c), and the right panel shows results based on the refined predictions in panel (f). The experimental data are significantly closer to the PO prediction than the nearly opposite MA prediction for both raw and generalization-refined versions of the predictions. In all panels, the shaded regions and error bars represent 95% CIs.

the greatest duration that we could reasonably use to examine feedforward motor output (see Materials and methods), spanning from movement onset to feedback response onset ($T_{RESP}$), and the other, more conservative, spanned from movement onset to target cue onset ($T_{ON}$). Using the prediction index values measured during both time intervals, we found that the observed 2-target trial force pattern is markedly more consistent with the PO prediction compared to the MA prediction ($T_{ON}$: prediction index = +0.77 ± 0.21 [95% C.I.], p=1.41 × $10^{-6}$, t(15) = 7.25; $T_{RESP}$: prediction index = +0.54 ± 0.16, p=4.78 × $10^{-6}$, t(15) = 6.52; 1-sample *t*-test; *Figure 2g*, left).

## Refinement of model predictions based on movement direction variability

Observation of participant hand trajectories, exemplified in *Figure 2d*, reveals that there is appreciable variability in movement directions on both 1-target FF trials and 2-target trials. This is important to consider because directional deviation from the intended target direction would lead to small but definite biases in the amount of adaptation for all three trained target locations (left/center/right), consequently biasing both the MA and PO predictions. Biases would occur, for example, because movements directed exactly in the 0° direction, towards the center target, would be associated with $FF_{CENTER}$ after training; however, off-target movements on either side of the 0° direction would be associated with an FF in-between $FF_{LATERAL}$ and $FF_{CENTER}$. Thus, a distribution of movement directions centered around the 0° direction would, on average, be associated with an FF in-between $FF_{LATERAL}$ and $FF_{CENTER}$, rather than an FF comprised solely $FF_{CENTER}$. This effect would be greater for movements farther off-target, resulting in a small but definite variability-dependent bias in the FF intended to be associated with a given target. We thus performed an additional experiment (Expt 1-GEN, n = 10) where we measured the adaptation associated with a range of 'off-target' movements to determine the size of this variability-induced bias (*Figure 2e*). Specifically, we employed a task design identical to Expt 1, with multi-FF training in the left/center/right target directions, but we replaced 2-target trials with 1-target error clamp trials and positioned them in directions that enabled a dense sampling for generalization of the trained adaptation (every 7.5° in-between the trained target directions). The resulting adaptation levels, shown in *Figure 2e*, show that the multi-FF environment generalizes nonlinearly across movement directions, with noticeable changes in adaptation around the trained target directions (we found an ~34% change 7.5° from the 0° trained target and 38–42% changes 7.5° from the ±30° trained targets).

The pattern of generalization observed in *Figure 2e* is well approximated ($R^2$ = 0.98) by a model based on the additive combination of Gaussians centered around the three trained target locations, in line with previous work examining the generalization of motor adaptation (*Hwang et al., 2003*; *Pearson et al., 2010*). We therefore used this model to estimate the average adaptation level over the distribution of movement directions associated with left/right 1-target FF trials for the MA prediction and 2-target trials for the PO prediction from Expt 1 (see Materials and methods). Each participant's expected level of adaptation was then used to scale the raw force profiles that comprise the predictions. We found that this refinement reduced the magnitude of the predicted peak forces by 20-25% for both the MA and PO models (see *Figure 2c vs. f*). Although the data were clearly more in line with the raw PO prediction than the raw MA prediction, there was still a noticeable mismatch between the PO prediction and the data (*Figure 2c, g*). However, taking generalization into account results in a refined PO prediction that is in even greater alignment with the data,

corresponding to prediction indices that are closer to +1 at both $T_{ON}$ and $T_{RESP}$ ($T_{ON}$: prediction index = +1.0 ± 0.4 [95% C.I.], p=1.58 × $10^{-5}$, t(15) = 5.85; *Figure 2g*, left; $T_{RESP}$: prediction index = +0.76 ± 0.3, p=1.72 × $10^{-5}$, t(15) = 5.80; 1-sample *t*-test; *Figure 2g*, right). Together these findings indicate that movements performed under uncertainty arise from an action plan that optimizes task performance.

## Obstacle avoidance can elucidate the mechanisms for motor planning under uncertainty

In Expt 1, the MA prediction that we tested against PO was based on the idea that the motor system performs a motor average at the level of motor output or force output, which could be viewed as a low-level version of MA. However, a higher-level version of MA is also possible, where the motor system may instead average the kinematics of the movements associated with the potential target locations, and then generate the motor output required for this averaged motion. In Expt 2, we test this higher-level version of MA against PO. To do so, we used an experimental approach based on the perturbation of movement kinematics rather than the perturbation of environmental force dynamics while assessing whether MA or PO can explain the intermediate movements that arise from motor planning in uncertain conditions. Specifically, we imposed kinematic perturbations by placing obstacles to block and deflect direct movements to targets or potential targets. We designed this experiment based on a subtle variation of an influential study that supported MA (*Stewart et al., 2014*). However, here we show that the original instantiation of this experiment, which we replicate (Expt 2a), fails to make readily dissociable predictions for MA vs. PO, but that a single modification of it (Expt 2b) leads to highly dissociable, and in fact essentially opposite, predictions for these two models of motor planning under uncertainty.

In Expts 2a (n = 8) and 2b (n = 26), participants made 20 cm cued-onset reaching arm movements (*Figure 3a*) towards either a single prespecified target or two potential targets, as in Expt 1. After a baseline period in which participants practiced these trial types without obstacles present (obstacle-free trials; *Figure 3b*), we began placing obstacles on some trials that would obstruct direct movements to a target or a potential target (*Figure 3c, d*). Please note that for simplicity the subsequent explanations in this section reference a left-side obstacle condition; however, the experimental design was balanced within participants to include an equal number of interspersed right-side obstacle trials. In Expt 2a, we used an obstacle with the same size and positioning as that in *Stewart et al., 2014*. This obstacle protruded 2 cm to the right and effectively infinitely far to the left of the direct path between the start position and the left target (i.e., the obstacle-obstructed target), and thus required *rightward* deflections for left 1-target trials, as shown in the upper panels of *Figure 3c, d*. By contrast, in Expt 2b, we used a pared-down version of this obstacle that protruded 2 cm to the right but 0 cm to the left of the direct path between the start position and left target, as shown in the lower panels of *Figure 3c, d*. This smaller obstacle permitted both leftward and rightward travel paths around it, but critically, promoted the less circuitous leftward deflections (see *Figure 3c*). Accordingly, all 8 participants in Expt 2a consistently veered rightward as required around the obstacle, and all 26 participants in Expt 2b consistently veered leftward around the obstacle, in line with the intent of the experiment design. Hand paths from example participants are shown in Figure 4a, and data indicating deflections observed for all participants are shown in Figure 5b, f.

The contrasting movement deflections produced by the different obstacles on 1-target trials in Expt 2a vs. Expt 2b result in contrasting predictions for the MA hypothesis in these two experiments. On 2-target trials, where uncertainty in motor planning is present, MA predicts that participants would average the motor plans associated with obstacle-obstructed left 1-target trials and the unobstructed right 1-target trials (*Equation 1* – baseline MA model prediction):

$$\hat{\mu}_2 = \frac{1}{2} \cdot \mu_{1A} + \frac{1}{2} \cdot \mu_{1B} \tag{1}$$

where $\hat{\mu}_2$ represents the predicted deflection, or safety margin, around the obstacle on 2-target trials, $\mu_{1A}$ and $\mu_{1B}$ represent the observed deflections, or safety margins, on obstacle-obstructed and unobstructed 1-target trials, respectively, and the $\frac{1}{2}$ coefficient for each term indicates that $\mu_{1A}$ and $\mu_{1B}$ are equally weighted in the motor average. Note that the baseline MA model is parameter-free

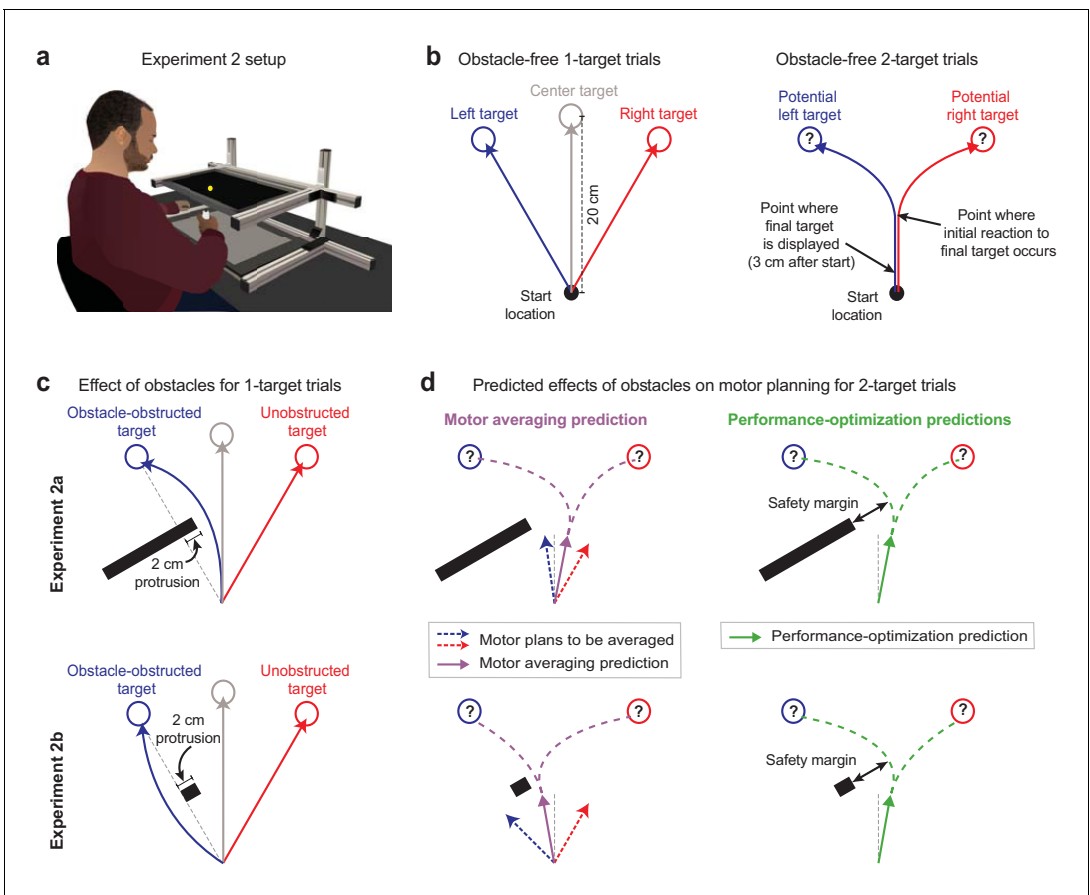

**Figure 3.** Obstacle avoidance paradigm. (a) Setup for the obstacle avoidance experiments, where virtual obstacles could obstruct movements to on-screen targets. (b) Illustration of obstacle-free 1-target and 2-target trials, similar to *Figure 1b*. (c) Diagram of left-side obstacle-present 1-target trials. The obstacle in Expt 2a was designed to elicit *rightward* movement deflections for left 1-target trials. The obstacle in Expt 2b, which included an identical protrusion towards the midline, was designed to promote a *leftward* deflection for left 1-target target trials. (d) Predictions during uncertainty. Left panels: motor averaging (MA; purple arrow) would average the plans for the obstacle-obstructed left and unobstructed right 1-target trial movements. It thus predicts initial movement directions on 2-target trials with opposite deflections from the midline for Expts 2a and 2b (*away* from the obstacle in Expt 2a, but *towards* the obstacle in Expt 2b, where the safety margin around the obstacle would be reduced). Right panels: performance optimization (PO; green) would promote task success by balancing the costs of safely avoiding the obstacle and rapid target acquisition. It thus predicts initial movement directions on 2-target trials to be consistently deflected *away* from the obstacle. Thus, PO and MA make similar predictions in Expt 2a, but opposite predictions in Expt 2b. Note that (c) and (d) depict a left-side obstacle condition, but that the experiments were balanced within participants, with an equal number of trials for left- and right-side obstacles. A right-side obstacle condition would lead to mirror-opposite predictions for both MA and PO.

in that both $\mu_{1A}$ and $\mu_{1B}$, which serve as inputs to the model, are experimentally measured on obstacle-obstructed and unobstructed 1-target trials. In Expt 2a, where movements on obstacle-obstructed left 1-target trials were deflected rightward, MA consequently predicts a rightward movement deflection on obstacle-present 2-target trials. And in Expt 2b, where movements on obstacle-obstructed left 1-target trials were deflected leftward, MA consequently predicts a leftward movement deflection on obstacle-present 2-target trials (see *Figure 3d*).

Importantly, the obstacles in Expts 2a and 2b were designed not only to induce opposite deflections on obstacle-obstructed 1-target trials, but also to display identical protrusions in the direction of the intermediate movements that occur during uncertainty. Thus a motor plan optimizing task performance for intermediate movements during uncertainty would not be differentially affected by these obstacles. Specifically, a model for PO would have two objectives on obstacle-present 2-target trials: (1) to reach the final target within the required timing criteria and (2) to avoid obstacle collision as these two objectives form the basis of task success (see Materials and methods). We therefore modeled the PO predictions as an average of the movement deflections that would arise if PO were

to independently prioritize each objective, expressing the PO prediction as an equal balance of the two motor costs associated with the determinants of task performance. Prioritization of movement timing, or rapid target acquisition, would predict a movement direction midway between the two potential targets (i.e., a net 0° deflection) as this would maximize the probability of successful target acquisition during uncertainty (*Haith et al., 2015a*). However, prioritization of obstacle avoidance would predict deflections that incorporate a safety margin around the obstacle that is proportional to an internal estimate of variability (*Hadjiosif and Smith, 2015*). We therefore estimated the sensorimotor system's safety margin for the obstacle avoidance priority on 2-target trials by scaling the margin observed on 1-target trials, $\mu_{1A}$, based on the relative variability observed on 2-target vs. 1-target trials, $\frac{\sigma_2}{\sigma_{1A}}$:

(*Equation 2a* – baseline PO model prediction):

$$\hat{\mu}_2 = \frac{1}{2} \cdot 0° + \frac{1}{2} \cdot \left( \mu_{1A} \cdot \frac{\sigma_2}{\sigma_{1A}} - 15° \right), \ if \ \left( \mu_{1A} \cdot \frac{\sigma_2}{\sigma_{1A}} - 15° \right) > 0° \qquad (2a)$$

(*Equation 2b* – baseline PO model prediction):

$$\hat{\mu}_2 = 0°, \ if \ \left( \mu_{1A} \cdot \frac{\sigma_2}{\sigma_{1A}} - 15° \right) \leq 0° \qquad (2b)$$

where $\hat{\mu}_2$ again represents the predicted deflection, or safety margin, on obstacle-present 2-target trials, the $\frac{1}{2}$ coefficient on each priority term confers equal weighting of the two priorities, the 0° term corresponds to full prioritization of movement timing, and the $\mu_{1A} \cdot \frac{\sigma_2}{\sigma_{1A}}$ term is the safety margin corresponding to full prioritization of obstacle avoidance – within that term, $\sigma_2$ and $\sigma_{1A}$ represent the observed motor variability on obstacle-present 2-target trials and obstacle-obstructed 1-target trials, respectively, and $\mu_{1A}$ represents the observed deflection, or safety margin, on obstacle-obstructed 1-target trials. Like the baseline MA model, the baseline PO model is parameter-free in that $\mu_{1A}$, $\sigma_2$, and $\sigma_{1A}$, which serve as inputs to the model, are experimentally measured on obstacle-obstructed 1-target trials and obstacle-present 2-target trials. Note that the population-averaged variability ratio $\frac{\sigma_2}{\sigma_{1A}}$ was calculated as the ratio of the mean of the individual participant values for $\sigma_2$ and $\sigma_{1A}$. Interestingly, we found population-averaged values of 1.02 and 1.00 for this ratio in Expts 2a and 2b, respectively, indicating that at the population level motor variability on obstacle-present 2-target trials and obstacle-obstructed 1-target trials were quite similar. Thus, inclusion of the factor $\frac{\sigma_2}{\sigma_{1A}}$ in the PO model does not meaningfully influence the prediction for the population-averaged 2-target trial movement direction, $\hat{\mu}_2$. However, although $\sigma_2$ and $\sigma_{1A}$ are similar on average, we found that they can vary considerably across individuals and even more so across movement directions within individuals. Oddly, individuals often displayed idiosyncratic but consistent differences in motor variability for different movement directions. Thus, differences in $\sigma_2$ and $\sigma_{1A}$ are not entirely surprising given the large differences in movement direction that can be observed on obstacle-obstructed 1-target trials and obstacle-present 2-target trials. Importantly, taking this idiosyratic direction-specific motor variability into account substantially enhances the ability to predict the safety margin observed in one movement direction from motor variability observed at another across individuals. Thus the factor $\frac{\sigma_2}{\sigma_{1A}}$ is important for predicting inter-individual differences in 2-target trial movement directions (see *Equations 5 and 6*).

The inclusion of the safety margin in motor planning for PO predicts deflections that skew *away* from the obstacle in both Expts 2a and 2b. In Expt 2a, MA, like PO, predicts rightward deflections that skew away from the obstacle on 2-target trials. However, in Expt 2b, MA predicts leftward deflections that skew towards the obstacle, whereas PO continues to predict rightward deflections that skew away from the obstacle. The 15° offset term that is subtracted from the $\mu_{1A} \cdot \frac{\sigma_2}{\sigma_{1A}}$ safety margin term is present because the obstacle protruded 15° away from the 0°, straight-ahead movement. Because of this 15° obstacle offset, the predicted movement deflection driven by full prioritization of obstacle avoidance would be 15° less than the $\mu_{1A} \cdot \frac{\sigma_2}{\sigma_{1A}}$ safety margin for that priority. Relatedly, if the $\mu_{1A} \cdot \frac{\sigma_2}{\sigma_{1A}}$ safety margin was 15° or smaller, corresponding to $\left( \mu_{1A} \cdot \frac{\sigma_2}{\sigma_{1A}} - 15° \right) \leq 0°$ in *Equation 2b*, then the 0° straight-ahead movement that fully satisfies the movement timing objective would already provide a safety margin at or above the level required for the obstacle avoidance objective,

removing the need to compromise between the two objectives because a 0° straight-ahead movement would simultensously optimize both. For simplicity, however, we drop this conditional secondary equation from the PO model variants presented below (*Equations 4–6*) as the condition required for using it is never met when applying these equations to the data from Expts 2a or 2b.

## Performance optimization predicts motor planning for obstacle avoidance

Like in Expt 1, we focused our analysis on the initial portion of motor output, measured as the initial movement direction, as this reflects feedforward motor planning. We calculated the movement direction as the direction of the hand at the midpoint of the movement (i.e., along the axis of the obstacle protrusion) relative to the direction of the hand at movement onset (but note that we obtained qualitatively similar results for all analyses when the movement direction was calculated 4 cm into the movement). In estimating each participant's mean 2-target trial movement direction, we combined the left-side and right-side obstacle data. To facilitate a balanced comparison between the obstacle-free and obstacle-present 2-target trial conditions, we randomly assigned each obstacle-free 2-target trial a left-side or right-side obstacle condition label and combined the data accordingly. Correspondingly, on obstacle-present 2-target trials, we found that participants displayed movement directions that were consistently biased *away* from the obstacle compared to obstacle-free movement directions in both Expt 2a (6.84 ± 4.51° vs. 0.32 ± 0.50° [mean ± 95% CI], p=9.40 × $10^{-3}$, t(7) = –3.04) and Expt 2b (2.7 ± 0.80° vs. 0.056 ± 0.29°, p=8.01 × $10^{-7}$, t(25) = –6.51; *paired t-test*). This is consistent with the predictions of PO, but at odds with the predictions of MA. Sample trajectory data are shown in *Figure 4a*, and population-averaged movement direction data are shown in *Figure 4b*.

As in Expt 1, we quantified the relative accuracy of the model predictions for MA and PO using a prediction index that attained a value of +1 if the experimental data matched the PO prediction, –1 if they matched the MA prediction, and 0 if they were midway between these two predictions (see Materials and methods). Correspondingly, we tested whether the data were significantly closer to the prediction of one model or the other by determining whether this prediction index was significantly above or below zero. We found that the experimentally observed movement directions on 2-target trials, which characterize motor planning in uncertain conditions, were far closer to the PO prediction than the MA prediction in Expt 2b, where gross differences in the PO and MA predictions were expected, with mean squared prediction errors of 4.24 ± 1.47 deg² (mean ± SEM) for the PO model and 163.50 ± 11.49 deg² for MA (see the 'baseline' MA and PO predictions in *Figure 4b*). Correspondingly, the mean prediction index was close to +1 (+0.96 ± 0.12; p=4.31 × $10^{-14}$, t(25) = 15.13; *t*-test). In Expt 2a, however, where gross differences between the MA and PO predictions were not expected, the mean obstacle-present 2-target trial data were fourfold closer to the PO prediction than the MA prediction, but the individual participant data was more variable than in Expt 2b. Furthermore, the span between the MA and PO predictions was smaller, providing less leverage to dissociate between them. These effects resulted in no significant difference between the models (prediction index = +0.59 ± 1.19; p=0.37, t(7) = 0.96; *t*-test; squared errors of 39.49 ± 32.18 deg² for PO and 73.04 ± 14.54 deg² for MA). Collectively, these findings suggest that during uncertainty, participants form an action plan that optimizes task performance instead of automatically averaging potential motor plans.

## Refinement of the motor averaging and performance optimization models

To allow the MA and PO models to make predictions without free parameters, we assumed equal weighting for both potential targets in the MA model, in line with previous work (*Chapman et al., 2010*; *Ghez et al., 1997*; *Gallivan et al., 2016*; *Van der Stigchel et al., 2006*), and analogously, for both objectives in the PO model (obstacle avoidance and rapid target acquisition), as specified in *Equations 1 and 2*. While this is reasonable, it is possible that this constraint might provide an explanation for why the baseline MA model could not accurately predict the experimental data. We therefore evaluated a refined version of the baseline MA model that removed this constraint by including a weighting parameter, $\alpha$, that permitted differential weighting of the obstacle obstructed and unobstructed targets, as shown in *Equation 3*:

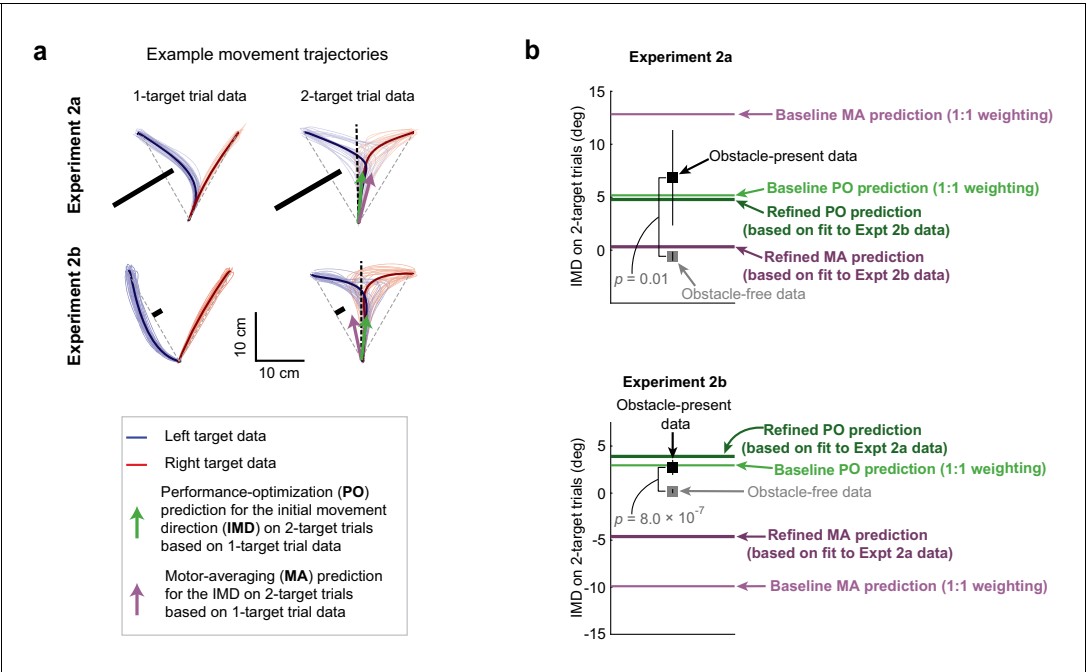

**Figure 4.** Obstacle avoidance patterns indicate performance optimization (PO) during uncertainty. (a) Hand paths showing the effect of obstacles on 1- and 2-target trials for sample participants in Expts 2a and 2b. Thin lines indicate a random sampling of trajectories on individual trials where the left (blue) or right (red) targets were cued, and bold traces indicate trial-averaged trajectories. On 1-targets, the obstacles induce substantial deflections for movements towards the obstacle-obstructed target, but in opposite directions for Expt 2a (n = 8) and Expt 2b (n = 26). Motor averaging (MA) PO predictions for the initial movement direction (IMD) on obstacle-present 2-target trials are displayed as purple and green arrows. These two predictions are similar in Expt 2a, but more distinct in Expt 2b. (b) Population-averaged IMDs on obstacle-free (gray squares) and obstacle-present (black squares) 2-target trials are shown alongside model predictions by MA (purple lines) and PO (green lines). Predictions from two versions of the MA model and two versions of the PO model are displayed. The 'baseline' predictions (which were used in panel a) were parameter-free and used equal weighting of the two potential motor plans for MA and of the two objectives for PO (see *Equations 1 and 2*). The 'refined' predictions each incorporated a single weighting parameter that determined the relative contributions of the two potential motor plans for MA and of the two objectives for PO (see *Equations 3 and 4*). Cross-validated predictions are shown for these 'refined' single-parameter models, where the weighting parameter applied to the Expt 2a data was determined from the Expt 2b data, and vice versa (see Materials and methods). Results indicate that on obstacle-present 2-target trials, IMDs were systematically biased away from the obstacle and were significantly larger than the IMDs observed on obstacle-free 2-target trials (compare black vs grey squares). Both the baseline and refined PO models predict the obstacle-present 2-target trial data significantly better than the baseline and refined MA models in Expt 2b, where gross differences in the model predictions were expected (compare black squares vs purple and green lines). In Expt 2a, where qualitative differences between the model predictions were not expected, we found that the baseline and refined PO models predicted the obstacle-present 2-target trial data only nominally better than the baseline and refined MA models. Error bars represent 95% CIs. The online version of this article includes the following figure supplement(s) for figure 4:

**Figure supplement 1.** Model sensitivity analysis.

*Equation 3* – refined MA model:

$$\hat{\mu}_2 = \alpha \cdot \mu_{1A} + (1 - \alpha) \cdot \mu_{1B} \tag{3}$$

Thus $\alpha = 1$ would indicate that all weight is assigned to the motor plan associated with the obstacle-obstructed target $\mu_{1A}$, $\alpha = 0$ would indicate that all weight is assigned to the motor plan associated with the unobstructed target $\mu_{1B}$, and $\alpha = \frac{1}{2}$ would indicate equal weighting, as in *Equation 1*. Note that the value for $\alpha$ and that for $\beta$ (defined below) was constrained to be between 0 and 1 to avoid any negative weighting. We found weighting coefficients of 0.26 and 0.74 for the obstacle-obstructed and unobstructed targets, respectively ($\alpha = 0.26$ when fitting to the Expt 2a data, and 0 and 1, $\alpha = 0$ when fitting to the Expt 2b data). However, to avoid the effects of overfitting, we evaluated these parameter estimates not based on their ability to explain the data on which they were fit, but instead, based on the ability of the parameter estimate obtained from one experiment's data to

predict the results of the other experiment. The findings from this cross-validation are discussed below.

Analogous to the refinement of the MA model, we assessed whether including differential weighting for the obstacle avoidance and rapid target acquisition objectives might improve the already-accurate predictions from the baseline PO model. We therefore also evaluated a refined version of the PO model that incorporated a weighting parameter, $\beta$, that allowed for differential weighting of these objectives:

*Equation 4* – refined PO model:

$$\hat{\mu}_2 = \beta \cdot 0° + (1 - \beta) \cdot \left( \mu_{1A} \cdot \frac{\sigma_2}{\sigma_{1A}} - 15° \right) \tag{4}$$

Thus $\beta = 1$ would indicate full prioritization of movement timing, $\beta = 0$ would indicate full prioritization of obstacle avoidance, and $\beta = \frac{1}{2}$ would indicate equal weighting of these priorities as in *Equation 2*. We found weighting coefficients for movement timing and obstacle avoidance of 0.66 and 0.34, respectively ($\beta = 0.66$), when fitting to the Expt 2a data, and 0.46 and 0.54 ($\beta = 0.46$) when fitting to the Expt 2b data. These weighting parameters suggest similar priority levels for movement timing and obstacle avoidance in Expts 2a and 2b. However, like for the refined MA model, we evaluated these parameter estimates not based on their ability to explain the data on which they were fit, but instead, based on the ability to predict the results of the other experiment (see *Figure 4b*).

Unsurprisingly, we found that when the parameters $\alpha$ and $\beta$ were fit onto the Expt 2a data, the resulting refined versions of the MA and PO models both exactly matched the mean data from Expt 2a, corresponding to mean squared errors of $37.03 \pm 14.31$ deg$^2$ across participants for both refined models. However, when the parameter estimates obtained from the Expt 2b data were cross-validated on the Expt 2a data, the prediction from the PO model increased the mean squared error by only 11%, whereas that from the MA model increased it by 115%. Accordingly, the experimental data from Expt 2a were noticeably closer to the cross-validated PO prediction than the cross-validated MA prediction, as shown in *Figure 4b*. The difference, however, was not significant (with a prediction index of $+1.11 \pm 1.98$; p=0.11, t(7) = 1.85; *t*-test; and mean squared errors of $79.52 \pm 53.71$ deg$^2$ vs. $40.93 \pm 33.89$ deg$^2$ for MA vs. PO). This finding was not surprising as Expt 2a was not expected to clearly dissociate the models.

In contrast, the data from Expt 2b, which was specifically designed to dissociate MA from PO, yielded clear results. First, due to the grossly opposite predictions for MA and PO in Expt 2b, the MA model was unable to exactly match the mean 2-target trial data when fit to the Expt 2b data, unlike the PO model (see *Figure 4b*). More importantly, when parameter estimates obtained from the same Expt 2a data were cross-validated on the Expt 2b data, the PO model prediction increased the mean squared error by 29%, whereas the MA model prediction increased it by 750% (squared error of $5.42 \pm 1.16$ deg$^2$ vs. $4.19 \pm 1.59$ deg$^2$ for PO compared to $58.10 \pm 7.20$ deg$^2$ vs. $6.78 \pm 2.67$ deg$^2$ for MA). Correspondingly, we found that the Expt 2b data were far better explained by the prediction from the PO model than by the prediction from the MA model (prediction index = $+0.74 \pm 0.19$; p=$5.79 \times 10^{-8}$, t(25) = 7.61), with mean squared errors that were 10-fold smaller for the PO model's prediction (squared errors of $58.10 \pm 7.20$ deg$^2$ vs. $5.42 \pm 1.16$ deg$^2$, for the MA and PO models, respectively).

These results show that the assumption of equal weighting was not the reason that the baseline MA model failed to outperform the baseline PO model, and they provide further support for the hypothesis that PO rather than MA explains motor planning under uncertain conditions. We note that to better understand each model's sensitivity to its weighting parameter we also examined predictions that were based on 1:2 and 2:1 weightings for the obstacle-obstructed and -unobstructed targets in the MA model, and for the movement timing and obstacle avoidance objectives in the PO model. However, we found that these predictions led to a similar pattern of results (see *Figure 4—figure supplement 1*).

## Performance optimization predicts individual differences in obstacle avoidance

After finding that the PO model accurately predicts the group-averaged data from Expt 2, we proceeded to examine whether it might also be able to explain inter-individual differences in motor

planning during uncertain conditions in Expts 2a and 2b. To study the ability of the PO model to explain inter-individual differences in motor planning, we created a version of it that combined the effects of using the group-averaged data and each individual participant's data, as shown below:

$$\hat{\mu}_{2_i} - \bar{\mu}_2 = \frac{1}{2}(k\mu_{1A_i} + (1-k)\bar{\mu}_{1A}) \cdot (k\sigma_{2_i} + (1-k)\bar{\sigma}_2) \cdot \left(k\frac{1}{\sigma_{1A_i}} + (1-k)\frac{1}{\bar{\sigma}_{1A}}\right) - K_0 \qquad (5)$$

Here, the relative weighting between the group-averaged and the individual participant data is given by a single parameter $k$, which we refer to as the individuation index. $k = 1$ corresponds to full individuation, whereas $k = 0$ corresponds to no individuation, that is, full weighting of the group-average data. The other variables in *Equation 5* are the same as those in *Equation 2*, but note that here the subscript $i$ refers to individualized data from participant $i$, and that a horizontal bar over a variable indicates the group-average value. Unlike *Equations 2 and 4*, this version of the PO model encodes the individual participant measurements of motor variability in $\sigma_{2_i}$ and $\sigma_{1A_i}$, which allows the model to account for individual differences in variability between 1- and 2-target trials that result from idiosyncratic spatial differences in movement direction. Note that the left side of *Equation 5* represents predicted inter-individual differences from the group-averaged obstacle-present 2-target trial movement direction because $\bar{\mu}_2$ is subtracted from $\hat{\mu}_{2_i}$, and analogously, that the right side represents inter-individual differences from the group-averaged model prediction because $K_0$, the group-averaged model prediction, is subtracted. Also note that the inclusion of this offset parameter, $K_0$, is necessary so that the model's ability to explain variance in the data is well-posed. The partial weighting of individual and group-averaged data that $k$ allows is often used for optimal estimation with noisy data, corresponding to the fact that the experimental estimates made for variables, like the mean or variability of the initial movement directions in our data, are affected by measurement noise to a far greater extent for individual participant measurements than for group averages.

Remarkably, when we fit this one-parameter-plus-offset model to the experimental data, we found that it was able to explain 87% of the variance for inter-individual differences in movement direction on 2-target trials in the Expt 2a dataset ($F(1,6) = 39.4$, p=7.60 $\times$ 10$^{-4}$, first bar in *Figure 5a*) and 57% of this variance in the Expt 2b dataset ($F(1,24) = 31.9$, p=8.21 $\times$ 10$^{-6}$, first bar in *Figure 5e*). This indicates that a majority of the inter-individual variability in the initial direction of intermediate movements planned under uncertain conditions in our experiments can be explained by a PO model that takes individual differences in 1-target trial safety margins, 1-target trial motor variability, and 2-target trial motor variability into account.

We next examined the extent to which the model's ability to predict inter-individual differences accrued from each of its input variables: 1-target trial safety margins ($\mu_{1A}$), 1-target trial motor variability ($\sigma_{1A}$), and 2-target trial motor variability ($\sigma_2$). To accomplish this, we compared a more complex three-parameter-plus-offset version of this model that included different individuation indices ($k_\mu$, $k_{\sigma_1}$, $k_{\sigma_2}$) for each input variable to two-parameter versions where the remaining individuation index was held at zero. This removed individuation for one of the three input variables and allowed us to calculate each input variable's partial R$^2$ value, which characterizes the contribution of that variable to the explained variance in a nested model. The three-parameter-plus-offset model is shown in *Equation 6*:

$$\hat{\mu}_{2_i} - \bar{\mu}_2 = \frac{1}{2}\left(k_\mu\mu_{1A_i} + (1-k_\mu)\bar{\mu}_{1A}\right) \cdot (k_{\sigma_2}\sigma_{2_i} + (1-k_{\sigma_2})\bar{\sigma}_2) \cdot \left(k_{\sigma_1}\frac{1}{\sigma_{1A_i}} + (1-k_{\sigma_1})\frac{1}{\bar{\sigma}_{1A}}\right) - K_0 \qquad (6)$$

Analysis of the three-parameter-plus-offset version of the PO model revealed two key observations. First, we found that the three-parameter-plus-offset model resulted in only marginal improvement in its ability to explain inter-individual differences in the data compared to the one-parameter-plus-offset model, suggesting that any true differences in individuation between the three input variables were relatively small (compare the first and second bars in *Figure 5a, e*). Second, we found that two of the three input variables – 1-target trial safety margins and 2-target trial motor variability – contributed significantly to explaining individual differences in the data, whereas 1-target trial motor variability did not. This was separately true for both Expt 2a (we found partial R$^2$ values of 0.90, 0.81, and 0.00 for $k_\mu$, $k_{\sigma_2}$, and $k_{\sigma_1}$, respectively; p=4.02 $\times$ 10$^{-3}$, 1.50 $\times$ 10$^{-2}$, and 1.000; $F(1,4) = $ 35.3, 16.7, 0.0) and Expt 2b (we found partial R$^2$ values of 0.32, 0.65, and 0.00 for $k_\mu$, $k_{\sigma_2}$, and $k_{\sigma_1}$,

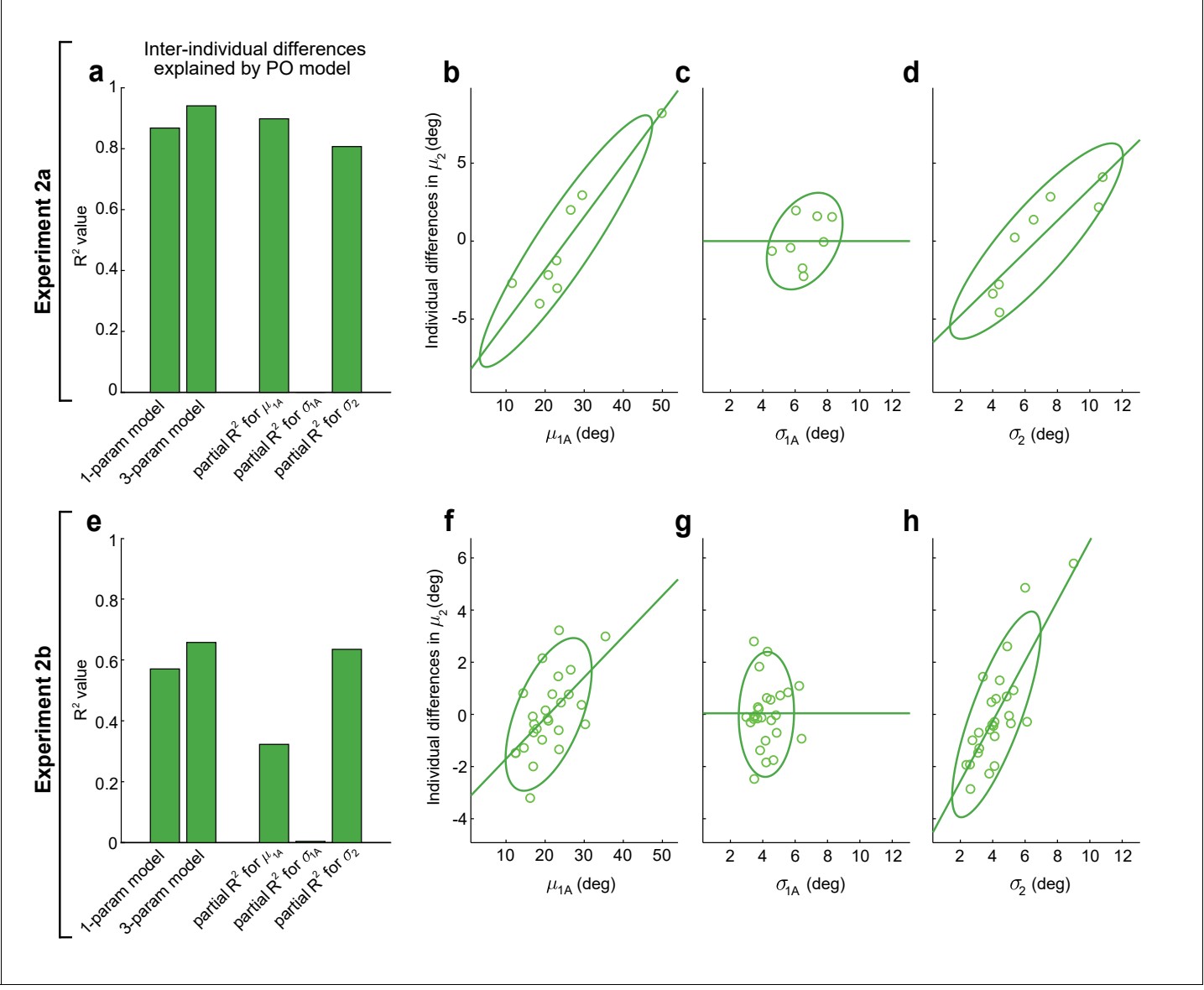

**Figure 5.** Performance optimization (PO) predicts individual differences in obstacle avoidance. (**a–d**) show analogous analyses to (**e–h**), for Expts 2a and 2b, respectively. (**a, e**) The first and second bars show the proportion of variance explained ($R^2$) for inter-individual differences in mean movement directions on obstacle-present 2-target trials, $\mu_2$, for the one-parameter-plus-offset and the three-parameter-plus-offset versions of the PO model (*Equations 5 and 6*). The remaining three bars show the proportion of variance explained (partial-$R^2$) for the three-parameter-plus-offset PO model attributable to each of the three input variables: the safety margin observed on obstacle-obstructed 1-target trials ($\mu_{1A}$), the motor variability observed on obstacle-obstructed 1-target trials ($\sigma_{1A}$), and the motor variability observed on obstacle-present 2-target trials ($\sigma_2$). The contributions of each input variable are illustrated in (**b–d**) for Expt 2a and (**f–h**) for Expt 2b, where the relationship is shown between each input variable and the individual differences in $\mu_2$ that remain after controlling for the effects of the other two input variables (see Materials and methods). In both Expts 2a and 2b, results indicate that $\mu_{1A}$ and $\sigma_2$ contribute significantly but $\sigma_{1A}$ does not. The lack of contribution from $\sigma_{1A}$ may be due to its small dynamic range seen in (**c**) and (**g**). Notably, the results in (**f**) are odds with motor averaging (MA), which predicts a negative relationship between $\mu_2$ and $\mu_{1A}$ in Expt 2b, as are the results in (**d**) and (**h**), where MA would predict no relationship between $\mu_2$ and $\sigma_2$. Green lines show linear fits to the data and ellipses show 95% CIs.

respectively; p=3.78 × 10⁻³, 3.18 × 10⁻⁶, and 0.763; *F*(1,22) = 10.5, 38.2, 0.09). Scatter plots of the relationships between each of the three input variables and the inter-individual differences that remain when the other two input variables are accounted for are shown in *Figure 5b–d* for Expt 2a and in *Figure 5f–h* for Expt 2b.

We were initially surprised to find that individuation using $k_{\sigma_1}$ did not meaningfully contribute to explaining the individual differences in 2-target trial movement directions. However, examination of the scatter plots in *Figure 5* reveals that the individual differences in 1-target trial motor variability were considerably smaller than those for the other two input variables. Thus, individual differences in 1-target trial motor variability, $\sigma_{1A}$, had considerably less leverage to explain individual differences in the 2-target trial movement direction, $\mu_2$, than the other input variables.

Importantly, these individual participant results provide further evidence at odds with the MA hypothesis. First, the clear positive relationship between the 2-target trial movement direction and the 1-target trial safety margin in Expt 2b (see *Figure 5e*) is opposite of the negative relationship that MA would predict (in line with the opposite predictions for MA and PO for the Expt 2b data illustrated in *Figure 4a, b*). Second, the clear positive relationships between the mean movement direction and the amount of motor variability on 2-target trials found in both in the Expt 2a data (*Figure 5d*) and the Expt 2b data (*Figure 5h*) are consistent with PO, but are at odds with MA, which predicts that there should be no relationship between these variables.

In summary, we find that PO provides an explanation not only for population-averaged motor planning for the intermediate movements that result from uncertainty about the target location, but also for a majority of the variance in inter-individual differences in this planning observed in Expts 2a and 2b. A detailed analysis reveals that individualized information about 1-target trial safety margins and 2-target trial motor variability both contribute significantly to the ability to explain these individual differences in motor planning.

## Discussion

We designed two novel experimental paradigms that powerfully dissociated between the hypotheses proposed to underlie motor planning under uncertainty: motor averaging (MA) and performance optimization (PO). In Expt 1, we designed an environment that physically perturbed motion in the direction of potential target locations off-course, and motion in the direction of intermediate movements oppositely off-course. Participants readily adapted to this composite environment on 1-target trials. Critically, on trials with two potential targets, participants displayed motor output strongly aligned with adaptive responses for intermediate movements, which was consistent with the PO prediction, but grossly opposite the MA prediction for motor planning under uncertainty (see *Figure 2*). An effort to refine the model predictions by taking the observed spread of movement directions into account resulted in even greater alignment between the observed motor output and the PO prediction.

A potential issue with Expt 1 was that, since motor output at the level of force production was measured, the MA model we considered was based on the averaging of force output. Thus the results cannot rule out the possibility of MA occurring for a higher level of motor output – that of motion planning. Therefore, we designed Expt 2 to dissociate between versions of the MA and PO models that were instead based on the kinematics of motion planning. Specifically, instead of perturbing environmental force dynamics, we used an obstacle avoidance task to perturb movement kinematics. In doing so, we replicated (Expt 2a) the paradigm from a well-known study that provided support for MA (*Stewart et al., 2014*) but did not qualitatively dissociate the predictions for MA and PO, and then made a seemingly minor modification (Expt 2b) that allowed for MA and PO to be powerfully dissociated. Specifically, in Expt 2a, we positioned an obstacle so that single-target movements would be skewed in a direction consistent with increasing the safety margin for intermediate movements during uncertainty. Thus, qualitatively, the PO prediction based on creating an appropriate safety margin for intermediate movements during uncertainty and the MA prediction based on averaging the motor plans for potential targets were in the same direction. Critically, however, in Expt 2b, we altered the obstacle from Expt 2a so that single-target movements would be skewed in a direction opposite to that needed for increasing the safety margin for intermediate movements during uncertain 2-target trials, resulting in opposite predictions for MA and PO (*Figure 3*). Experimentally, we found that the presence of this obstacle altered motor output during 2-target trials by consistently increasing the safety margin for intermediate movements in accord with the PO prediction, but grossly opposite from the MA prediction. An attempt to refine the MA model to allow for different weightings of the motor plans associated with the obstacle-obstructed and -unobstructed targets did not lead successfully lead to predictions consistent with the experimental

data. On the other hand, the PO model accurately predicted the population-averaged changes in motor output for both experiments as well as a remarkable ~65–94% of the variance for individual differences between participants. In line with previous work investigating the determinants of task optimization (*Trommershäuser et al., 2003*; *Hamilton and Wolpert, 2002*; *Harris and Wolpert, 1998*; *Todorov and Jordan, 2002*), this finding suggests that internal estimates of variability and uncertainty can be critical for motor planning. Collectively, our results provide clear evidence that humans generate a motor plan that reflects optimization of task performance rather than averaging over multiple potential plans.

## Previous work suggesting motor averaging for motor planning under uncertainty

Our finding that PO underlies movements executed during uncertainty challenges the long-standing idea that the intermediate movements executed during uncertain conditions reflect MA. However, many studies suggesting that either sensory or motor representations of movement plans are averaged did not dissociate these possibilities from selecting a single plan that optimizes motor performance. For example, the *Stewart et al., 2014* study that we replicate in Expt 2a attempted to dissociate whether motor output might reflect an average of sensory or motor representations of movement plans (i.e., sensory averaging vs. MA) by using obstacles to perturb these motor plans, but did not dissociate either sensory averaging or MA from PO.

Another study, by *Gallivan et al., 2017*, applied a visuomotor rotation to movements towards one potential target to perturb motor planning without affecting the sensory representation of the target. This resulted in initial movement directions that were altered during 2-target trials in accordance with the perturbed motor plans, and thus in line with the prediction for MA rather than sensory averaging. However, PO predicts a movement direction identical to that predicted by MA: since the imposed visuomotor rotation shifts the final hand position associated with acquisition of the potential target to which it was applied, a corresponding shift in the movement direction that accounts for this visuomotor rotation would optimize performance by minimizing the cost of corrective movements following disclosure of the final target location. Therefore, this experimental manipulation, like that in *Stewart et al., 2014* and a number of other studies (*Chapman et al., 2010*; *Stewart et al., 2014*; *Gallivan et al., 2011*; *Gallivan et al., 2016*; *Gallivan and Chapman, 2014*; *Stewart et al., 2013*; *Chou et al., 1999*; *Van der Stigchel et al., 2006*), fails to dissociate MA from PO.

## Previous work on performance optimization for motor planning under uncertainty

*Haith et al., 2015a* examined motor planning under uncertainty using a timed-response reaching task where the target suddenly shifted on a fraction (30%) of trials (150–550 ms) before movement initiation. The authors observed intermediate movements when the target shift was modest (±45°), but direct movements towards either the original or shifted target position when the shift was large (±135°). The authors argued that because intermediate movements were not observed under conditions in which they would impair task performance, that motor planning under uncertainty generally reflects PO. This interpretation is somewhat problematic, however. In this task, like in the current study, the goal location was uncertain when initially presented. However, the final target was presented far enough before movement onset that this uncertainty was no longer present during the movement itself, as evidenced by the direct-to-target motion observed when the target location was shifted by ±135°. Therefore, the intermediate movements observed when the target location shifted by ±45° are unlikely to reflect motor planning under uncertain conditions. Instead, these intermediate movements likely arose from a motor decision to supplement the plan elicited by the initial target presentation with a corrective augmentation when the plan for this augmentation was certain. The results thus provide beautiful evidence for the ability of the motor system to flexibly modulate the correction of existing motor plans, ranging from complete inhibition to conservative augmentation, when new information becomes available, but provide little information about the mechanisms for motor planning under uncertain conditions.

The challenge of examining motor planning under uncertainty has persisted even for studies that made deliberate attempts at dissociating MA from PO. For example, in *Wong and Haith, 2017*,

motor planning under uncertainty was examined under conditions where participants were required to reach towards one of two potential targets with a movement completion time that was small enough to preclude movements with corrective adjustments from being successful. The results showed that under these conditions participants frequently abandoned intermediate movements that subsequently require corrective adjustments on 2-target trials in favor of direct movements towards one of the potential targets. These findings indicate that MA is not obligatory, and that conditions can be created where intermediate movements themselves are abandoned. This stands in contrast to the present study, in which intermediate movements were allowed, yet were found to be inconsistent with MA, suggesting that MA is unlikely to occur at all. In addition, it is likely that the task conditions used in Wong and Haith simply induced participants to make a strategic decision to guess the final target location and aim at that guess. While this would indeed improve performance and could therefore be considered a type of performance optimization, such strategic decision making does not provide information about the implicit neural processing involved in programming the motor output for the intermediate movements that are normally planned under uncertain conditions.

In another study, performed by *Nashed et al., 2017*, motor planning under uncertainty was studied using a task that was based on grip force control. Specifically, participants grasped an object, capable of measuring grip force, and made reaching movements while environmental dynamics were experimentally manipulated with robotically applied load forces that affected the required grip force. This task design resembles Expt 1 from the current study in which different environmental dynamics were used to perturb movements towards the left/right vs. center targets in order to dissociate MA from PO. However, grip force control is known to be substantially more sensitive to the variability in environmental dynamics than to the mean dynamics (*Hadjiosif and Smith, 2015*), yet this study examined the MA and PO hypotheses for motor planning under uncertainty using predictions that were based entirely on the mean environmental dynamics. Unfortunately, this study did not provide information about motor variability or its effect on required grip forces, obscuring the driver for the observed changes in grip force. Thus, it is unclear how this study can elucidate the mechanisms that drive motor planning under uncertainty.

## Implicit and explicit contributions to motor planning under uncertainty

An important consideration for the present results is that sensorimotor control engages both implicit and explicit adaptive processes to generate motor output (*Mazzoni and Krakauer, 2006*). Because motor output reflects combined contributions of these processes, determining their individual contributions can be difficult. In particular, the experiments in the present study used environmental perturbations to induce adaptive changes in motor output, but these changes may have been partially driven by explicit strategies, and thus the extent to which the motor output measured on 2-target trials reflects implicit vs. explicit feedforward motor planning requires further investigation. One method for examining implicit motor planning during goal uncertainty might take inspiration from recent work showing that in visuomotor rotation tasks, restricting the amount of time available to prepare a movement appears to limit explicit strategization from contributing to the motor response (*Haith et al., 2015b*; *Huberdeau et al., 2019*; *Leow et al., 2017*; *Fernandez-Ruiz et al., 2011*). Future work could dissociate the effects of MA and PO on intermediate movements in uncertain conditions at movement preparation times short enough to isolate implicit motor planning.

We note that *Gallivan et al., 2017* attempted to control for the effects of explicit strategies by (1) applying the perturbation gradually, so that it might escape conscious awareness, and (2) enforcing a 325 ms preparation time. Intermediate movements persisted under these conditions, suggesting that intermediate movements during goal uncertainty may indeed be driven by implicit processes. However, it is difficult to be certain whether explicit strategy use was, in fact, effectively suppressed as the study did not assess whether participants were indeed unaware of the perturbation. On one hand, the single study we are aware of that assessed the extent to which reduced movement preparation times can suppress strategization during motor learning, showed that participants can successfully execute a re-aiming strategy when specifically instructed to do so, even down to movement preparation times as small as 150 ms. On the other hand, when a specific strategy was not instructed, the same study found that a 250 ms preparation time yielded performance that was indistinguishable from implicit learning measured using an aim-report paradigm with a long (1000 ms) preparation time, suggesting that active strategization was effectively suppressed. It should be noted, however, that the difference between 250 and 325ms is considerable and its

effect unclear, and also that any claim of full suppression in the 250ms data would depend on a negative statistical comparison (note further that the 250ms and 325ms nominal preparation times quoted in these papers are larger than the true preparation times as the quoted times include both display latencies and data-acquisition latencies; neither of these were reported in Gallivan et al., and only the former was reported in Leow et al. (27.6 ± 1.8 ms)).

## Neural representations of motor planning under uncertainty

Studies suggesting motor or sensory averaging have been motivated by reports of simultaneous deliberation of competing potential goals in sensorimotor brain areas (i.e., parallel planning) (*Chapman et al., 2010*; *Stewart et al., 2014*; *Gallivan et al., 2011*; *Stewart et al., 2013*; *Gallivan et al., 2015*). These studies argue that the motor plans prepared in parallel are averaged, resulting in the intermediate movements observed when uncertainty is present during motor planning. The neural evidence for parallel planning is based on studies of delay period activity associated with motor planning when multiple potential targets were present (*Cisek and Kalaska, 2005*; *Cisek, 2007*; *Coallier et al., 2015*; *Pastor-Bernier and Cisek, 2011*). However, because this activity was measured using single-electrode recordings, only a small number of cells could be recorded simultaneously, and thus data from same-type trials were aggregated to make population-based estimates of the planned motion. The results suggested that this aggregate delay period activity was tuned to both potential targets in sensorimotor areas, in particular dorsal premotor cortex (PMd) and parietal reach region, in monkeys. However, the observed tuning could arise from simultaneous parallel planning for both potential targets, as the authors argued, or from planning-related activity associated with one target on some trials and with the alternate target on other trials.

Recently, *Dekleva et al., 2018* studied parallel planning using an electrode array to record simultaneously from 100 to 160 neurons in PMd, allowing planning-related neural activity to be examined for individual movements when two potential targets were present. Critically, they found that neural activity during the delay period encoded motor plans directed to either one target or the other on individual trials, rather than simultaneous parallel plans for both potential targets. While these results cannot rule out parallel planning in other brain areas, they call into question the evidence for parallel planning from previous studies that relied on trial-aggregated data. Taken together with the results from the current study, it is plausible that during uncertainty the brain forms a single motor plan that optimizes task performance rather than multiple parallel motor plans for MA.

In summary, the current findings indicate that motor planning during uncertain conditions does not proceed from averaging parallel motor plans, but instead, incurs the creation of a motor plan that optimizes task performance given knowledge of the current environment. These findings are compatible with the current neurophysiological data and offer a mechanistic framework for understanding motor planning in the nervous system.

# Materials and methods

## Participants

26 participants (25 right-handed; 15 females; age range 18–42) performed the multi-force-field (multi-FF) adaptation experiments, with 16 participants in Expt 1 and 10 participants in Expt 1-GEN. 34 participants (32 right-handed; 18 females; age range 18–33) performed the obstacle avoidance experiments, with 8 participants in Expt 2a and 26 participants in Expt 2b. Participants were assigned to experiments based on when they responded to advertisements and their availability for scheduling. When different experiments were running concurrently, participants were randomly assigned for participation. The sample sizes used for each experiment were determined based on pilot data and existing literature (*Stewart et al., 2014*; *Smith et al., 2006*; *Sing et al., 2009*). All participants used their right hands to perform the experiments, were naïve to the purpose of the experiments, and were without known neurological impairment. The study protocol was approved by the Harvard University Institutional Review Board (protocol number: IRB16-2128), and all participants provided written informed consent.

### Experiment protocols

## Apparatus for multi-FF adaptation experiments (Expt 1 and Expt 1-GEN)

Participants were instructed to grasp the handle of a two-joint robotic manipulandum with their right hands and make rapid 20 cm point-to-point reaching arm movements in the horizontal plane to either a single prespecified target (1-target trials) or two potential targets (2-target trials). All visual information, including veridical feedback of hand motion provided in the form of a white 3-mm-diameter cursor, was displayed on a vertically oriented LCD computer monitor (refresh rate of 75 Hz). Participants were positioned such that their midlines were aligned with the middle of the monitor, and their right arms were always supported with a ceiling-mounted sling. The manipulandum measured hand position, velocity, and force, and its motors were used to dynamically apply prescribed force patterns to the hand, all of which were updated at a sampling rate of 200 Hz.

## Targets and feedback (Expt 1 and Expt 1-GEN)

On 1-target trials, the target was located at a left (+30°), right (−30°), or center (0°) eccentricity from the midline, and on 2-target trials, a pair of potential targets always appeared at the left and right target locations. Participants were instructed to initiate a trial by moving the cursor to a start position (green filled-in circle, 10-mm-diameter), after which the target or targets (yellow hollow circles, 10-mm-diameter) were presented. 1000 ms after target presentation, an auditory go cue signaled participants to initiate a movement. Movement onset was subsequently determined online as the time when the hand velocity exceeded 5 cm/s or the time when the hand traveled 3 cm from the start position (whichever occurred first). Participants were required to initiate movements after the onset of the go cue, but no later than 425 ms after the go cue finished playing. If movement onset was detected outside these bounds, a message that either read 'Too Soon!' or 'Too Late!' was displayed above the start position, and was accompanied with a unique sound tone. Furthermore, because pilot data showed that participants may sporadically stop and initiate a discrete reach to the final target immediately following its disclosure on 2-target trials, we also required participants to maintain their velocity throughout the first half of every movement (i.e., while the displacement was less than 10 cm). Specifically, during each trial, if the instantaneous maximum velocity declined more than 33% during the first half of the movement, we played a unique buzzer tone. If any of these requirements were not fulfilled, the trial was discarded.

For the 2-target go-before-you-know trials, the final target (randomly selected on each trial) was filled in with yellow, and the distractor target was simultaneously extinguished once a 3-cm displacement between the hand and start position was achieved. For consistency, on 1-target trials, the target also filled in with yellow at the same point in the movement. After the hand reached the final target, we provided performance feedback based on the movement time, determined as the time interval between movement onset, as defined above, and movement offset, defined as the first time-point when the hand was both within 6 mm of the final target, and the hand speed in the subsequent 300 ms period was below a threshold of 6.35 cm/s. Following movement offset, visual and auditory feedback were presented by changing the fill color of the target and playing a sound, depending on whether the movement time was faster than (red fill-in color, buzzer tone), within (green fill-in color, chirp tone), or slower than (blue fill-in color, buzzer tone) the required interval. This movement time interval was based on thresholds that were adjusted online per participant as described below. After feedback was delivered, the robotic manipulandum guided participants' hands back to the start position. Participants completed blocks of trials throughout the experiments but were allotted 1 min rest breaks in-between each block (see *Training schedules [Expt 1 and Expt 1-GEN]* below).

## Movement time thresholds

We used data-driven updating for specifying the movement time thresholds. The 'too-slow' threshold was set at the 70th percentile of the movement times associated with the last 18 trials of the same type (1-target or 2-target). Thus, separate thresholds were maintained for 1-target vs. 2-target trials. The 1-target trial movement time threshold ranged from 225 to 585 ms in Expt 1 and 225 to 608 ms in Expt 1-GEN (on average across participants). In Expt 1, the 2-target trial movement time thresholds ranged from 225 to 655 ms. We used this 70th percentile updating to individualize the movement time thresholds for different participants because we found that, in pilot data, individuals

with a large fraction of 'too-slow' feedback sometimes exhibited erratic, seemingly exploratory behavior on 2-target trials, and abandoned intermediate movements. Individuals with a smaller fraction of 'too-slow' feedback, however, did not. Intermediate movements are of fundamental interest for the examination of motor planning during uncertainty as these movements have been previously taken to indicate deliberation between potential goals, but unfortunately, abandonment of intermediate movements has remained an issue in studies with standard implementations of go-before-you-know tasks, consequently leading to striking data exclusion criteria (e.g., removal of 10-40% of participants in previous work; *Gallivan et al., 2017*; *Nashed et al., 2017*; *Stewart et al., 2013*; *Stewart et al., 2014*). Individualized movement time thresholding allowed us to evoke intermediate movement behavior consistently within and across all participants.

## Multi-FF environment

To dissociate between performance optimization (PO) of intermediate movements during goal uncertainty and motor averaging (MA) of actions associated with each potential goal presented during uncertainty, we differentially perturbed the 1-target motor plans in a manner that resulted in distinct predictions for each hypothesis in Expt 1. To create such a perturbation, we used the robot motors to produce a dynamic environment comprising multiple curl FFs that acted on the manipulandum to perturb hand motion off-course in opposite directions depending on the cued target direction during 1-target trials in Expt 1 and Expt 1-GEN. This multi-FF environment levied velocity-dependent FFs that were proportional in magnitude and directionally orthogonal to the velocity of hand motion,

$$\begin{bmatrix} F_x \\ F_y \end{bmatrix} = k \cdot \begin{bmatrix} 0 & -B \\ -B & 0 \end{bmatrix} \cdot \begin{bmatrix} \dot{x} \\ \dot{y} \end{bmatrix} \begin{bmatrix} F_x \\ F_y \end{bmatrix}$$

where $\dot{x}$ and $\dot{y}$ denote the hand velocities, $B = 15 \frac{Ns}{m}$ denotes the velocity-dependent gain, and $k = \pm 1$ denotes a binary switch variable that was set to opposite values for center target trials vs. left target or right target trials for each participant. Setting $k = +1$ results in clockwise (CW) FFs, whereas setting $k = -1$ results in counterclockwise (CCW) FFs. We balanced the directions of the applied FFs across participants so that half experienced the multi-FF environment with $k = +1$ for left target or right target trials (e.g., see FF$_{LATERAL}$ in *Figure 1c*) and $k = -1$ for center target trials (e.g., see FF$_{CENTER}$ in *Figure 1c*), and the other half experienced the multi-FF environment with $k = -1$ for left target or right target trials and $k = +1$ center target trials. Data associated with each target were then combined from each subgroup.

## Error clamp and partial error clamp trials

Because actions made during reaching movements may result from both feedforward motor planning and online feedback corrections to movement errors, we used error clamp trials to restrict deviations from the straight-line path towards the target during 1-target trials. We implemented these error clamp trials as a highly stiff (6000 N/m), viscous (250 Ns/m) one-dimensional spring and damper system in the direction orthogonal to the straight-line path between the initial hand position and the cued target. In line with previous work, these error clamp trials effectively eliminated movement errors (average maximum absolute deviation, <1.9 mm) and allowed for a high-accuracy measurement of feedforward participant-produced forces patterns (*Scheidt et al., 2000*; *Sing et al., 2009*).

Unlike 1-target trials, feedback corrections are to be expected during 2-target trials if task success is to be achieved because participants must reflexively correct their movements following divulgence of the ultimate target. We thus devised a variant of the error clamp, which we term the partial error clamp, to measure the initial segment of force output during 2-target trials that reflects feedforward motor planning *before* these feedback corrections occur. As the initial motion on these 2-target trials was directed towards the center target, we aligned the partial error clamps to the straight-line path connected to the center target location, but we smoothly transitioned the hand from the highly stiff and viscous environment into a null environment (i.e., the robot motors were disabled) after a 11-cm distance threshold. Note that the 11-cm point was selected based on an analysis of pilot data to determine the onset of feedback corrections to the final target. These partial error clamp trials allowed us to measure feedforward force patterns early in the movement, while

motor errors were minimized (average maximum absolute deviation, <2.3 mm), but still permitted participants to carry out feedback corrections later in the movement for final target acquisition. Post-hoc surveys indicated that 5/16 participants noticed partial error clamps, whereas 6/16 participants noticed the standard error clamps.

## Training schedules (Expt 1 and Expt 1-GEN)

We divided the experiment into baseline, training, and test epochs with a total of 1305 outward-reaching movements. The experiment began with the baseline epoch, which consisted of nine blocks. The first three blocks comprised 120 null 1-target trials that familiarized participants with the basic experimental setup and feedback structure described above. The next four blocks comprised 200 2-target trials, the first 50 of which were null trials, and the remaining 150 were 80% null trials and 20% partial error clamp trials. The last three blocks of the baseline period reacquainted participants with 1-target trials before the training period started and comprised 85 1-target trials, in which 80% were null trials and 20% were error clamp trials. The force patterns measured on error clamp and partial error clamp trials throughout this epoch were used as a baseline for estimating learning-related changes in force patterns during subsequent blocks. Note that in this epoch all blocks that comprised 1-target trials probed each target direction in equal amounts.

The baseline epoch was followed by the training epoch, which was separated into seven blocks and comprised exclusively of 1-target trials. The purpose of the training epoch was to elicit robust adaptation to the multi-FF environment we designed. However, because this environment differentially perturbs movements to the center target compared to the lateral (left/right) targets (see *Multi-FF environment*), interference effects for adaptation to the FFs associated with the lateral targets would arise from center target training. Importantly, whereas the source of interference for left/right target training is singular, center target training would suffer interference effects from the FFs associated with both lateral targets. Interference effects would limit the overall level of adaptation that can be achieved, and because these effects are non-uniformly distributed across the trained target directions, straightforward application of the multi-FF environment may lead to dissimilar levels of adaptation across target directions. These issues would consequently limit the statistical power in dissociating MA from PO and would bias their predictions. We therefore sought to create a training schedule that would avoid this bias and elicit similar levels of adaptation to the FFs associated with each target. To do so, we tested various training schedules in a pilot study that was guided by simulations from a linear state-space model with local motor primitives for predicting the effects of generalization in our composite environment. We correspondingly employed the FF training schedule we found successful, which included 500 training trials (with 80% FF trials and 20% error clamp trials), and twice the number of center target trials compared to the number of left or right target trials. Note that this distribution of target directions (1:2:1 for left, center, and right target directions, respectively) carried into the next epoch.

After the training epoch, participants completed the test epoch, which was separated into five blocks. The purpose of the test epoch was to probe participants' initial force patterns during 2-target trials while approximately maintaining the level of adaptation to the multi-FF environment achieved for 1-target trials during the training epoch. We thus included 400 trials in this epoch, of which 300 were 1-target FF trials, 50 were 1-target error clamp trials, and 50 were 2-target partial error clamp trials. Note that during all epochs error clamps and partial error clamps were interspersed in a pattern that was random (frequency of one in five during the baseline and training epochs, and one in four during the test epoch) but which avoided consecutive error clamps or partial error clamps trials to prevent adaptation decay.

The training schedule of Expt 1-GEN was analogous to that of Expt 1, but 2-target trials were replaced with 1-target trials that were positioned at one of nine different directions (from −30° to 30° every 7.5°). Thus the training epoch of Expt 1-GEN was identical to that of Expt 1, but during the baseline epoch, participants reached towards each of the nine targets, presented in random order, for an equal number of trials. Baseline force patterns associated with each target were probed with error clamps on 20% of trials after the third baseline block as in Expt 1. During the test epoch, the 2-target partial error clamp trials from Expt 1 were replaced with 1-target error clamp trials that probed generalization to the 'off-targets' positioned in-between the trained target directions (i.e., these trials probed the targets located at ±22.5°, ±15°, and ±7.5°).

## Apparatus for obstacle avoidance experiments (Expts 2a and 2b)

Participants were instructed to grasp a lightweight plastic handle that sheathed a digital stylus and make reaching movements with their right hands in the horizontal plane. We instructed participants to slide the handle across the surface of a tablet capable of recording hand position at 200 Hz with a resolution of 0.01 mm. All visual stimuli, including targets, obstacles, and a real-time cursor showing hand position, were displayed on a horizontally oriented LCD computer monitor (with a screen refresh rate of 120 Hz and a motion display latency of ~25 ms) that was mounted above the tablet at the shoulder level and therefore obstructed view of the hand. Participants were positioned such that their midlines were aligned with the middle of the monitor and tablet.

## Design of obstacles (Expts 2a and 2b)

In Expts 2a and 2b, participants made reaching movements using the 1-target and 2-target trial configurations from Expt 1, but on some trials, we presented visual obstacles that we instructed participants to avoid. As illustrated in *Figure 3c*, the obstacle in Expt 2a was rectangular in shape (width 1 cm and length 12 cm) and was oriented so that its long axis was perpendicular to the vector between the start position and the target. Moreover, it was positioned midway between the start location and the target, and from this location, protruded 2 cm towards the midline and 10 cm away from it so that movements around the obstacle would be consistently deflected towards the midline. In Expt 2b, we modified the obstacle from Expt 2a by clipping off the 10 cm away-from-midline protrusion, so that it still protruded 2 cm towards the midline, but now 0 cm away from it (see *Figure 3c*, bottom panel). This modification promoted deflections that, relative to the target directly blocked by the obstacle, were away from the midline and opposite in direction to those in Expt 2a. This dichotomy allowed us to dissociate the MA and PO hypotheses for motor planning under uncertainty.

## Trial types and feedback (Expts 2a and 2b)

Expts 2a and 2b included obstacle-free and obstacle-present 1- and 2-target trials. Visual stimuli and feedback were identical to those of Expt 1. Participants were instructed to rapidly reach the final target but while avoiding a virtual obstacle, if present. A custom collision-detection algorithm was created so that if a collision with the obstacle was detected, a message that read 'You hit the obstacle!' was displayed, a buzzer tone was played, and the trial was disqualified from movement time reward feedback. Otherwise, the movement time feedback and the individualized thresholding procedure was identical to Expt 1, but with separate thresholds maintained for 2-target trials, obstacle-obstructed 1-target trials, and all remaining 1-target trials (i.e., all obstacle-free 1-target trials and all obstacle-present 1-target trials in which the obstacle did not directly block the target). We did not maintain separate thresholds for obstacle-free vs. obstacle-present 2-target trials because pilot studies indicated that the differences in movement completion times were small. The 2-target trial movement time thresholds ranged from 225 to 645 ms in Expt 2a and 225 to 598 ms in Expt 2b (on average across participants). The obstacle-obstructed 1-target trial movement time threshold ranged from 225 to 450 ms in Expt 2a and 225 to 360 ms in Expt 2b. The movement time threshold for the remaining 1-target trials ranged from 225 to 380 ms in Expt 2a and 225 to 382 ms in Expt 2b. After participants reached the final target, they were instructed to move the handle back to the starting position to begin the next trial.

Note that all participants in both experiments completed an equal number of trials in which the obstacle was positioned between the start target and the left target (left-side obstacle condition) and between the start target and right target (right-side obstacle condition). The experiments were therefore balanced within participants to cancel out any target-specific effects that might lead to biases in movement direction. *Figure 3* displays the MA and PO predictions based on a left-side obstacle condition, but for a right-side obstacle condition, the geometry of the predictions would simply be left/right mirror-reversed. Data shown in *Figure 4b* and *Figure 5* correspondingly combine the left-side and right-side obstacle conditions for each participant.

## Training schedules (Expts 2a and 2b)

Expts 2a and 2b had identical training schedules, and thus the experiments only differed in the obstacle geometry, as described above, and shown in *Figure 3*. Both experiments were divided into

a baseline and test epoch with a total of 760 outward-reaching movements. The baseline epoch included nine blocks (460 trials total) and was designed to familiarize participants with the basic task and feedback structure before obstacle-present 2-target trials were presented in the test epoch. The first two blocks comprised 120 obstacle-free 1-target trials, and the next two blocks comprised 100 obstacle-present 1-target trials. Of the five blocks that followed, which included a total of 240 trials, three comprised obstacle-free 1- and 2-target trials (with 40% 1-target trials and 60% 2-target trials, 180 trials total) and the remaining two blocks comprised solely obstacle-present 1-target trials (60 trials total). Note that across all baseline blocks, the 1-target trials probed each target direction in equal amounts.

The test epoch included six blocks (300 trials total) and was designed to probe motor planning for obstacle-present 2-target trials so that predictions for MA and PO could be compared. All blocks in this epoch comprised solely obstacle-present trials, with 60% obstacle-present 1-target trials and 40% obstacle-present 2-target trials. Of the obstacle-present 1-target trials, we probed movements towards the obstacle-obstructed target more often (50% of trials) because our analyses were more sensitive to movements towards this target compared to movements towards the center or unobstructed targets. The remaining 50% of obstacle-present 1-target trials probed the center and unobstructed targets in equal amounts. Note that, in both the baseline and test epochs, left-side and right-side obstacle conditions were separated into different blocks, and the ordering of these blocks was balanced across participants.

## Analysis

### Outlier analysis

No participants were excluded from any dataset. Individual movements that did not comply with the task requirements, outlined in *Targets and feedback (Expt 1 and Expt 1-GEN)*, were not eligible for analysis (<3% of trials in Expt 1, <2% of trials in Expt 1-GEN, <1% of trials in Expt 2a, and <2% of trials in Expt 2b). In addition, we discarded a small fraction of highly atypical movements based on two key features. For all experiments, we required that the movement time was between 225 ms and 2000 ms (<1% of trials in Expt 1, <1% of trials in Expt 1-GEN, <1% of trials in Expt 2a, and <1% of trials in Expt 2b) and for Expt 1 and Expt 1-GEN, we also required that peak velocity was between 0.2 m/s and 1 m/s (<1% of trials in Expt 1 and <1% of trials in Expt 1-GEN). Note that only movements performed after the familiarization blocks in each experiment were used for analysis, and for those movements, these criteria collectively resulted in the omission of <3% of trials in Expt 1, <2% of trials in Expt 1-GEN, <1% of trials in Expt 2a, and <2% of trials in Expt 2b.

### Analysis of force patterns in Expt 1

We examined the lateral force profiles participants produced that were orthogonal to the cued target direction for 1-target EC trials and to the center target direction for 2-target partial error clamp trials, corresponding to the axis of the imposed perturbations (see *Multi-FF environment*). We aligned all force profiles to the onset of the target cue ($T_{ON}$, 40–50 ms after movement onset) and used the population-averaged force profiles measured during the test period, after participants became acclimated to the multi-FF environment, to construct the MA and PO predictions. Specifically, we constructed the MA prediction by averaging the force profiles associated with the left and right targets, and then constructed the PO prediction by directly using the force profiles associated with the center target (*Figure 2c*). Since we sought to isolate the feedforward component of the data and predictions, before feedback responses to the target cue occurred, we analyzed all force profiles until the minimum time (across participants) that differences in force output on left and right cued 2-target partial error clamp trials were significantly different from zero ($T_{RESP}$, ~150 ms after $T_{ON}$). In addition, because the imposed FF environment was velocity-dependent, and adaptive responses to velocity-dependent dynamics are known to be scaled by movement velocity from one trial to the next (*Joiner et al., 2011*), we normalized each force profile by the velocity-dependent level of ideal compensation.

For a simple determination of how participants compensated for the multi-FF environment we imposed, we characterized the adaptive response on 1-target trials with an adaptation coefficient, calculated as the slope from a linear regression of the baseline-subtracted force profiles participants made during error clamp trials onto the ideal compensatory force (*Smith et al., 2006*; *Sing et al.,*

*2009*; *Hadjiosif and Smith, 2015*). For trials that were associated with the FF perturbations imposed during movements towards the center target, we defined the adaptation coefficient so that full FF compensation would yield an adaptation coefficient of +1. For trials that were associated with the FF perturbations imposed during movements towards the left or right target, we defined the adaptation coefficient so that full FF compensation would yield an AC of −1.

To quantify the similarity between the 2-target trial force data and the predictions, as shown in *Figure 2g*, we devised a prediction index that results in a value of +1 if the 2-target trial data is perfectly similar to the PO prediction, −1 if it is perfectly similar to the MA prediction, and 0 would if the data were halfway between both predictions,

$$PI = \frac{\mu_2 - \mu_C}{\mu_D}$$

$$\mu_C = \frac{\mu_{PO} + \mu_{MA}}{2}, \; \mu_D = \frac{\mu_{PO} - \mu_{MA}}{2}$$

where $\mu_C$ and $\mu_D$ correspond to the common and differential modes, respectively, of predicted mean force levels based on the PO ($\mu_{PO}$) and MA ($\mu_{MA}$) models, and $\mu_2$ corresponds to the mean force level of the 2-target trial data. In Expt 1, we calculated the prediction index over two intervals: one spanned movement onset until $T_{ON}$, and the other spanned movement onset until $T_{RESP}$.

### Refinement of predictions based on generalization of adaptive responses (Expt 1 and Expt 1-GEN)

Due to non-trivial variability in motor output, participants occasionally deviated from the intended target direction on 1-target trials and from the center target direction on 2-target trials. Directional deviations consequently bias both the MA and PO predictions since adjacent targets were associated with different FFs in the composite environment we designed. To account for this variability-induced effect and refine our predictions, we first measured how the multi-FF environment generalizes to nine different movement directions in Expt 1-GEN (see *Training schedules [Expt 1 and Expt 1-GEN]*). We then estimated the generalization of adaptation throughout our composite environment by fitting the population-averaged adaptation coefficients (from the test period) associated with every probed target direction onto a model that was based on the additive combination of Gaussians centered around the trained target directions (+30°/0°/−30°),

$$g(\theta) = -A_1 e^{-\frac{(\theta-30)^2}{2\sigma^2}} + A_2 e^{-\frac{\theta^2}{2\sigma^2}} - A_1 e^{-\frac{(\theta+30)^2}{2\sigma^2}} + A_0$$

This equation describes the amount of generalization, $g$, as a function of the probed target direction, $\theta$. There are four free parameters: $\sigma$ is the width of each Gaussian, $A_1$ and $A_2$ are the heights of the Gaussians associated with left/right target training and center target training, respectively, and $A_0$ is an offset. This model uses equal width Gaussians for the adaptation at each target, which is reasonable since FFs levied in each target direction were structurally identical. We allowed different heights to be associated with the left/right vs. center target directions since movements towards the center target were perturbed twice as many times as movements towards the left or right targets (see *Training schedules [Expt 1 and Expt 1-GEN]*).

We used the model for $g(\theta)$ to determine how the force profiles associated with the MA and PO predictions from Expt 1 would be refined given each participant's distribution of movement directions. We determined the movement direction as the direction of the hand when it was 10 cm away from the start position relative to the direction of the hand at movement onset (and note that the distributions of movement directions shown in *Figure 2e* were based on random samples from the pooled aggregate of participant data). We used each participant's distribution of movement directions on left and right 1-target FF trials (from the test epoch) to refine the MA predictions, and the movement directions on null 2-target trials to refine the PO predictions. We applied the model for $g(\theta)$ to predict the pattern of generalization up to ±42.5° (∼2.5$\sigma$) away from the center target direction to obtain an estimate of generalization that encompassed the space of movement directions explored by participants. We then convolved the resulting model-estimated generalization function with each participant's distribution of movement directions from Expt 1 to determine each participant's distribution of adaptive responses conditioned on their observed directional variability. Using

each participant's expected value of adaptation from this distribution, we scaled the force profiles that previously formed what we refer to as the raw MA and PO predictions, and ultimately determined what we refer to as the refined predictions plotted in *Figure 2f*.

## Motor averaging and performance optimization models in Expts 2a and 2b

As outlined in the Results, the MA model posits that on 2-target trials, where uncertainty about the final goal location is present, individuals exhibit a motor plan that reflects an average of the motor plans associated with each potential goal,

$$\hat{\mu}_2 = \alpha \cdot \mu_{1A} + (1 - \alpha) \cdot \mu_{1B}$$

where $\hat{\mu}_2$ represents the predicted mean deflection, or safety margin, around the obstacle on 2-target trials, and $\mu_{1A}$ and $\mu_{1B}$ represent each participant's observed mean movement deflection, or safety margin, on obstacle-obstructed and unobstructed 1-target trials, respectively. Note that the hat symbol indicates a prediction, whereas variables without a hat symbol indicate an observed variable; subscript values of 1 and 2 indicate 1- and 2-target trial types, respectively; subscript characters of $A$ and $B$ indicate trial types associated with the obstacle-obstructed and -unobstructed targets, respectively. Also note that $\alpha$ is a weighting parameter that controls the influence of the motor plans associated with the obstacle-obstructed and -unobstructed target. Thus $\alpha = 1$ indicates that participants assigned all weight to the motor plan associated with the obstacle-obstructed target and $\alpha = 0$ indicates that participants assigned all weight to the motor plan associated with the unobstructed target. In line with canonical MA theories (*Nashed et al., 2017*; *Gallivan et al., 2017*; *Stewart et al., 2013*; *Stewart et al., 2014*), we assumed $\alpha = \frac{1}{2}$ for the baseline MA model presented in *Equation 2* and for the corresponding predictions presented in *Figure 4b*.

The PO model posits that on 2-target trials, individuals exhibit a motor plan that attempts to achieve task success given knowledge of the environment. Thus, on obstacle-present 2-target trials, PO of intermediate movements would have two objectives: (1) to reach the final target within the required timing criteria and (2) to avoid obstacle collision because as outlined in *Trial types and feedback (Expts 2a and 2b)*, these two objectives determined reward on each obstacle-present trial. We modeled the PO prediction as a combination of the predicted movement directions that would arise if PO were to independently optimize each objective. Optimization of (1) movement timing for task performance would lead to movement directions in the center target direction (0°) as this movement direction maximizes the probability of successful target acquisition during uncertainty (*Hudson et al., 2007*; *Haith et al., 2015a*). On the other hand, to determine the movement direction predicted if individuals were to optimize (2) obstacle avoidance, we exploited the observation that the motor system linearly modulates the size of safety margins for actions based on internal estimates of variability (*Hadjiosif and Smith, 2015*). Accordingly, we determined the expected safety margin around the obstacle during uncertainty by scaling the magnitude of safety margins observed during goal *certainty* by the change in variability that is induced during goal *uncertainty*. Thus the PO model took the following form:

$$\hat{\mu}_2 = \beta \cdot 0° + (1 - \beta) \cdot \left( \mu_{1A} \cdot \frac{\sigma_2}{\sigma_{1A}} - 15° \right)$$

where $\hat{\mu}_2$ again represents the predicted mean deflection, or safety margin, on obstacle-present 2-target trials, $\sigma_2$ and $\sigma_{1A}$ represent the observed variabilities on obstacle-present 2-target trials and obstacle-obstructed 1-target trials, respectively, and $\mu_{1A}$ again represents the observed mean deflection, or safety margin, on obstacle-obstructed 1-target trials. The variability ratio $\frac{\sigma_2}{\sigma_{1A}}$ was calculated as the ratio of the mean of the individual participant values for $\sigma_2$ and $\sigma_{1A}$. However, because the population-averaged values for $\sigma_2$ and $\sigma_{1A}$ were nearly identical, the variability ratio $\frac{\sigma_2}{\sigma_{1A}}$ was close to one (1.02 and 1.00 in Expts 2a and 2b, respectively) and thus had little effect on the output, $\hat{\mu}_2$, of the population-averaged version of the PO model shown above. However, the inclusion of this variability ratio in the participant-individualized version of the model (see below) is critical because on an individual participant level, variability differences between obstacle-obstructed 1-target trial and obstacle-present 2-target trial data, although largely idiosyncratic, do occur. And when taken into account, they afford a substantially improved ability to predict individual differences in the size of

safety margins on both trial types. The 0° term in the model reflects the movement direction midway between the potential targets that would optimize movement timing, the $\mu_{1A} \cdot \frac{\sigma_2}{\sigma_{1A}}$ variability-adjusted safety margin term reflects optimization of obstacle avoidance, and the 15° offset term that is subtracted from the $\mu_{1A} \cdot \frac{\sigma_2}{\sigma_{1A}}$ safety margin term is present because the obstacle is offset 15° from the 0° direction midway between the potential targets. Analogous to the MA model, we included a parameter $\beta$ to weigh the priority levels for the optimization of movement timing and optimization of obstacle avoidance objectives. $\beta = 1$ would indicate full prioritization of movement timing, $\beta = 0$ would indicate full prioritization of obstacle avoidance, and $\beta = 1 - \beta = \frac{1}{2}$ would indicate equal weighting of these priorities. We used this equal weighting for the baseline PO model presented in *Equation 2* and for the corresponding predictions presented in *Figure 4b*, corresponding to an equal balance of the two motor costs associated with the determinants of task performance. However, note that it is possible that 1-target trials display a safety margin that may already be somewhat of a balance between motor costs. If such weighting on obstacle-obstructed 1-target trials were present, the weighting coefficient $\beta$ for 2-target trials could be more precisely interpreted as the relative weighting between the motor costs on 2-target compared to 1-target trials. That said, we would expect the weighting of movement timing that $\beta$ provides to be considerably stronger for 2-target trials compared to 1-target trials as the time pressure to reach the correct target on 2-target trials should be considerably greater given that the final target location is revealed only after movement onset.

We further note that because the movement direction that optimizes movement timing on 2-target trials (0°) would not be directly obstructed by the obstacle, it already affords some amount of safety margin around the obstacle. Thus the movement direction that optimizes movement timing would also correspond to the movement direction that optimizes obstacle avoidance if the expected safety margin during uncertainty is sufficiently small (i.e., $\beta = 1$). However, if the safety margin provided by the 0° movement is not large enough to optimize obstacle avoidance, then the movement direction that optimizes both determinants of task success would indeed be based on a combination of the 0° movement direction associated with optimization of movement timing and a unique, positively valued movement direction (i.e., skewed away from the obstacle) that optimizes obstacle avoidance based on variability. This feature of the PO model would lead to predictions that evolve in a piece-wise linear fashion with respect to the variability-scaled 1-target trial safety margin, as outlined in *Equation 1*, but we note that the safety margin estimate associated with optimization of obstacle avoidance went beyond the 0° direction for the participants in both Expts 2a and 2b. Thus, for simplicity, we withheld displaying this condition from *Equations 4–6*.

The baseline MA and PO models assumed equal weightings for both potential targets in the baseline MA model, and analogously, for both determinants of task performance in the baseline PO model (obstacle avoidance and rapid target acquisition). In addition, however, we also assessed how differential weighting for the obstacle-obstructed and -unobstructed targets might affect the MA prediction, and analogously, how differential weighting for obstacle avoidance and rapid target acquisition might affect the PO prediction. To accomplish this, we fitted the MA and PO models to both the Expt 2a and 2b participant data to obtain two separate estimates of $\alpha$ and $\beta$ that reflect participants' own subjective valuations for the motor plans, or constituents, that comprise each model. Importantly, the values for $\alpha$ and $\beta$ were restricted to within the range of 0 and 1 to prevent negative weighting. To estimate $\alpha$ in the MA model, we regressed the difference between the individual participant 2-target trial and the unobstructed 1-target trial movement directions onto the difference between the individual participant obstacle-obstructed and -unobstructed 1-target trial movement directions. To estimate $\beta$ in the PO model, we regressed the individual participant 2-target trial movement directions onto the obstacle-obstructed 1-target trial movement directions.

The parameter estimates for $\alpha$ and $\beta$ were used to form refined predictions for both the Expt 2a and 2b data (shown in *Figure 4b*). However, for each experiment's dataset, we evaluated refined model predictions that were based on parameter estimates obtained from the other experiment's dataset. This cross-validation procedure allowed us to form predictions while avoiding the possibility of overfitting. However, we note that the predictions drawn for the sample participant data in *Figure 4a* were based on the baseline, not refined, MA and PO models. As an aside, we note that the trial-averaged trajectories displayed in this panel were determined by linearly interpolating the

x-positions of the hand path onto a vector of y-positions every 0.254 mm to align the hand path measurements across trials, allowing us to control for variations in hand velocity.

Like in Expt 1, we determined the consistency of the data to the MA and PO model predictions with the prediction index (see *Analysis of force patterns in Expt 1*), where the common and differential modes ($\mu_C$ and $\mu_D$) corresponded to the predicted population-averaged movement directions for obstacle-present 2-target trials based on the respective PO and MA models being compared, and the data ($\mu_2$) corresponded to the observed population-averaged movement directions on obstacle-present 2-target trials. In addition, we evaluated each model's performance by calculating the squared error between a given model's population-averaged prediction and each participant's mean observation. We calculated the squared error in this manner to report both the mean-squared-error and an associated SEM.

Because we found that the PO model was able to accurately predict the population-averaged obstacle-present 2-target trial movement direction (see *Figure 4b*), we explored how well the model might be able to predict individual differences in movement direction as well. Extending the baseline PO model (in which $\beta = \frac{1}{2}$) to predict each participant's movement direction yields the following,

$$\hat{\mu}_{2_i} = \frac{1}{2} \cdot 0° + \frac{1}{2} \cdot \left( \mu_{1A_i} \cdot \frac{\sigma_{2_i}}{\sigma_{1A_i}} - 15° \right)$$

where $\hat{\mu}_{2_i}$ represents the predicted mean deflection, or safety margin, on obstacle-present 2-target trials for each participant $i$, $\sigma_{2_i}$ and $\sigma_{1A_i}$ represent each participant's observed variabilities on obstacle-present 2-target trials and obstacle-obstructed 1-target trials, respectively, and $\mu_{1A_i}$ represents each participant's observed mean deflection, or safety margin, on obstacle-obstructed 1-target trials. A linear combination of this participant-individualized model, with a model based on population-averaged input variables, yields the following hybrid,

$$\hat{\mu}_{2_i} - \bar{\mu}_2 = \frac{1}{2}(k\mu_{1A_i} + (1-k)\bar{\mu}_{1A}) \cdot (k\sigma_{2_i} + (1-k)\bar{\sigma}_2) \cdot \left( k\frac{1}{\sigma_{1A_i}} + (1-k)\frac{1}{\bar{\sigma}_{1A}} \right) - K_0$$

where the individuation index $k$ indicates the relative weighting between the population-averaged input variables (variables with hat symbols) and the individualized input variables, and the parameter $K_0$ is the offset. We fit this one-parameter-plus-offset model separately onto the Expt 2a and 2b datasets to determine the amount of variance in inter-individual differences on obstacle-present 2-target trials that can be explained by a PO model that accounts for individual differences in $\mu_{1A}$, $\sigma_2$, and $\sigma_{1A}$. This fitting procedure, like all other others, was based on minimization of model errors in a least-squares sense.

Remarkably, we found that the one-parameter-plus-offset model was able to explain a majority of the variance in individual differences, with estimated values of 0.69 and 0.38 for $k$ in Expts 2a and 2b, respectively. We next examined the extent to which this ability was driven by each of its input variables: $\mu_{1A}$, $\sigma_2$, and $\sigma_{1A}$. We thus devised a simple extension of the one-parameter-plus-offset model that took the following form,

$$\hat{\mu}_{2_i} - \bar{\mu}_2 = \frac{1}{2}\left(k_\mu \mu_{1A_i} + (1-k_\mu)\bar{\mu}_{1A}\right) \cdot (k_{\sigma_2}\sigma_{2_i} + (1-k_{\sigma_2})\bar{\sigma}_2) \cdot \left( k_{\sigma_1}\frac{1}{\sigma_{1A_i}} + (1-k_{\sigma_1})\frac{1}{\bar{\sigma}_{1A}} \right) - K_0$$

This three-parameter-plus-offset model differed from the one-parameter-plus-offset variant in that each of the input variables ($\mu_{1A}$, $\sigma_2$, and $\sigma_{1A}$) was assigned a unique individuation index in $k_\mu$, $k_{\sigma_1}$, and $k_{\sigma_2}$. When fitting this model, we found that estimates for $k_\mu$, $k_{\sigma_2}$, and $k_{\sigma_1}$ led to values of 0.64, and 0.51, and 0 in Expt 2a, and values of 0.30, 0.45, and 0.04 in Expt 2b, respectively. Using these parameter estimates, we then compared the amount of variance in inter-individual differences on 2-target trials explained by this model to nested forms of the model in which one of the input variable's individuation effects was held constant, allowing us to calculate that variable's associated partial $R^2$ value (see *Figure 5a, e*). Correspondingly, we also visualized these contributions by plotting the relationship between each of the three input variables and the inter-individual differences in obstacle-present 2-target trials that remain when the other two input variables are held constant (see *Figure 5b–d, f–h*). Specifically, for each input variable, we determined predictions from a nested model in which the individuation indices associated with the other two input variables were set to 0, which effectively removes their ability to drive model predictions based on individual

differences between participants, and instead drives the model predictions based entirely on population means for those variables. We then added each of these reduced model predictions with the residuals accrued from the full three-parameter-plus-offset model to isolate the effects of the reduced model for each variable of interest. As indicated in the equations above, these predictions were subsequently mean-subtracted, resulting in predictions for individual differences on 2-target trials expressly linked to each respective input variable.

### Statistical tests

$t$-tests were used for most statistical comparisons as indicated in the Results section. A one-sided $t$-test was used to compare obstacle-present versus baseline obstacle-free 2-target trial movement direction data in Expt 2 (*Figure 4b*), and the remaining $t$-tests were two-sided. When we assessed nested forms of the full PO model in Expt 2 (see *Equations 5 and 6*), we used $F$-tests. In all statistical tests, the significance level (alpha) was set to 0.01, and normality assumptions for the tests were verified with Kolmogorov–Smirnov tests. Experiments were programmed in C++ or MATLAB, and all data were analyzed in MATLAB (RRID:SCR_001622).

## Acknowledgements

We thank Ashleigh Victoria Conroy-Zugel for help with experiments. We also thank J Ryan Morehead for creating the experimental setup illustrations (*Figures 1a* and *3a*). This work was supported by a grant from the National Institute on Aging (R01 AG041878) to MAS.

## Additional information

### Funding

| Funder | Grant reference number | Author |
| --- | --- | --- |
| National Institute on Aging | R01 AG041878 | Maurice A Smith |
| National Institute of Neurological Disorders and Stroke | R01 NS105839 | Maurice A Smith |

The funders had no role in study design, data collection and interpretation, or the decision to submit the work for publication.

### Author contributions

Laith Alhussein, Conceptualization, Data curation, Formal analysis, Methodology, Writing - original draft, Writing - review and editing; Maurice A Smith, Conceptualization, Resources, Formal analysis, Funding acquisition, Methodology, Writing - original draft, Writing - review and editing

### Author ORCIDs

Laith Alhussein ![iD] https://orcid.org/0000-0003-0411-1683
Maurice A Smith ![iD] https://orcid.org/0000-0003-4214-1277

### Ethics

Human subjects: The study protocol was approved by the Harvard University Institutional Review Board (protocol number: IRB16-2128), and all participants provided written informed consent.

### Decision letter and Author response

Decision letter https://doi.org/10.7554/eLife.67019.sa1
Author response https://doi.org/10.7554/eLife.67019.sa2

## Additional files

### Supplementary files

- Transparent reporting form

## Data availability

All data acquired for this study, used to analyze and generate all Figures have been deposited in OSF. https://doi.org/10.17605/OSF.IO/S8A2W.

The following dataset was generated:

| Author(s) | Year | Dataset title | Dataset URL | Database and Identifier |
|---|---|---|---|---|
| Alhussein L, Smith MA | 2021 | Motor planning under uncertainty | https://doi.org/10.17605/OSF.IO/S8A2W | Open Science Framework, 10.17605/OSF.IO/S8A2W |

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
