## [Decision Letter]

**Acceptance summary:**

In this paper, Alhussein and Smith set out to determine whether motor planning under uncertainty (when the goal is not fixed before the start of the movement) results in motor averaging (average between the two possible motor plans) or in performance optimization (one movement that maximizes the probability of successfully reaching to one of the two targets). Building on previous work, the authors provide two elegant behavioural studies, providing convincing evidence for the performance optimization and against the motor averaging hypothesis.

**Decision letter after peer review:**

Thank you for submitting your article "Motor planning under uncertainty" for consideration by *eLife*. Your article has been reviewed by 3 peer reviewers, including Jörn Diedrichsen as the Reviewing Editor and Reviewer #1 and the evaluation has been overseen by Tamar Makin as the Senior Editor.

Essential Revisions:

In this paper, Alhussein and Smith set out to determine whether motor planning under uncertainty (when the goal is not fixed before the start of the movement) results in motor averaging (average between the two possible motor plans) or in performance optimization (one movement that maximizes the probability of successfully reaching to one of the two targets).

In the first experiment the authors associate a pair of targets with FF_A and a target in the middle with FF_B. The authors show that under uncertainty between the two targets, participants exhibit FF_B, rather than FF_A, which would be predicted when motor commands would be averaged.

Experiment 2 then introduces an obstacle for one of the targets. In Experiment 2b, the predictions of motor averaging and performance optimization are pitched against each other. The results of both experiments are better explained by performance optimization, and are inconsistent with the motor averaging hypothesis.

Summary of main points in the editorial discussion:

1. Novelty: As the authors point out, there has been some amount of evidence against the motor averaging hypothesis (e.g. Haith et al., 2015; Wong and Haith, 2017; Dekleva et al., 2018). In an effort to place their study, the authors relatively briefly dismiss these studies (line 85-92) as somewhat inconclusive, but the rationale behind the limitation of the previous studies, and how the current studies solve these, remains unclear (see reviewer 3 for detailed comments). On the other hand, all reviewers agreed that the paper is likely the cleanest and most comprehensive test of the motor averaging hypothesis to date, and that there is value in these results, even though the underlying conclusions may not be entirely "novel". We would recommend a more balanced discussion of previous results, focussing less on perceived shortcomings of previous studies, but focussing more clearly on the value these current experiments have to add.

2. Some parts of the analysis and presentation were not clear. This concerns especially the correction of motor variability, applied to Experiment 2 (see comments from reviewer 1 and 2). First, both reviewers questioned the validity and justification of this factor. If one wants to know how much safety margin the motor system uses to plan a trajectory given a specific estimated amount of motor variability, why would the system use a different amount in a case where that optimal trajectory was a compromise between two targets? The additional variability in the trajectories caused by planning uncertainty should not be taken into account.

Secondly, it is not clear whether and to what degree the main conclusion depends on this correction factor – if it does, the value of second study would be substantially diminished in the eyes of the reviewers. Overall, the presentation of this section needs to be made much more accessible – or some analyses, if not absolutely crucial for the main conclusions to hold – should be removed.

The revision should also include a point-to-point response to the concerns raised by the individual reviewers as shown below.

*Reviewer #2 (Recommendations for the authors):*

1. Line 230: during this epoch ◊ which epoch?

2. Line 288: it is unclear what is being compared here. What is the contrast?

3. Figure 5: it should be made clearer that top row is linked to exp2a and bottom one to 2b. Unclear that the labels on the left are for the three panels on that line.

4. Figure 5c and 5f: the authors might want to use robust correlations to get rid of the influence of outliers rather that doing separate analyses with and without outliers.

5. Line 456: the authors should avoid using the word "predict" as they are only using correlation and not out-of-sample predictions.

6. Line 461-468: the motivation behind this analysis is unclear to me. In addition, the concept of polarity seems to be ill-defined (dichotomization of a continuous variable). What are these tests comparing ? How are these comparisons linked to the MA or PO predictions.

7. Line 516-518: robust correlations should be preferred and the authors should display these correlations if they want to make any conclusion on this basis.

8. Line 520-522: I don't understand how this conclusion follows from the correlation analysis above.

9. Line 523-530: Mu2i, MuHat2i, MuBar1A are used without any explanations of the Hat, Bar, presence or absence of "i" as index, etc.

10. Line 551-553: this is not a prediction, it is a fit….

*Reviewer #3 (Recommendations for the authors):*

The correlation results w/r/t variability (Figure 5) are an impressive extension of the concept of optimal planning and a rather nice discovery. This finding could perhaps be related, in the discussion or introduction, to bayes optimal incorporation of variability in motor planning under risk (e.g. Trommershäuser et al., 2003).

A visual schematic of the PO model could be helpful?

---

## [Author Response]

1. Novelty: As the authors point out, there has been some amount of evidence against the motor averaging hypothesis (e.g. Haith et al., 2015; Wong and Haith, 2017; Dekleva et al., 2018). In an effort to place their study, the authors relatively briefly dismiss these studies (line 85-92) as somewhat inconclusive, but the rationale behind the limitation of the previous studies, and how the current studies solves these, remains unclear (see reviewer 3 for detailed comments). On the other hand, all reviewers agreed that the paper is likely the cleanest and most comprehensive test of the motor averaging hypothesis to date, and that there is value in these results, even though the underlying conclusions may not be entirely "novel". We would recommend a more balanced discussion of previous results, focussing less on perceived shortcomings of previous studies, but focussing more clearly on the value these current experiments have to add.

We thank the editor and the reviewers for the kind words here. We have now provided a more balanced discussion of the previous results as suggested. In particular, we have made substantial revisions to the fourth (lines 90-94), fifth (line 98), sixth (lines 117-118), and seventh (lines 127-128) paragraphs in the revised introduction, and the fifth (lines 598-614), sixth (lines 622-623), seventh (lines 632-634), and ninth (lines 654-661) paragraph in the revised discussion to address this. Additionally, regarding reviewer 3’s specific comments about implicit versus explicit planning alluded to above, we agree that they are on the mark, and we have made revisions that address them – see the point-by-point response to reviewer 3’s comments below for details.

2. Some parts of the analysis and presentation were not clear. This concerns especially the correction of motor variability, applied to Experiment 2 (see comments from reviewer 1 and 2). First, both reviewers questioned the validity and justification of this factor. If one wants to know how much safety margin the motor system uses to plan a trajectory given a specific estimated amount of motor variability, why would the system use a different amount in a case where that optimal trajectory was a compromise between two targets? The additional variability in the trajectories caused by planning uncertainty should not be taken into account.

We apologize for the misunderstanding here, but there is no additional variability in the trajectories for 2-target trials compared to 1-target trials that would correspond to evidence for the existence of what the editor calls ‘planning variability’ (and what reviewer 1 calls ‘decision’ variability). Specifically, the average initial movement direction variability, as measured by SD, is so similar that it’s nominally within 1% for 2-target vs 1-target trials (4.80±0.37° vs 4.78±0.24°). Note that a plot quantifying 2-target trial variability vs 1-target trial variability in detail (Author response image 2) is included in the point-by-point response to Reviewer 1 below.

Secondly, it is not clear whether and to what degree the main conclusion depend on this correction factor – if it does, the value of second study would be substantially diminished in the eyes of the reviewers.

In line with the fact that the population-averaged variability is essentially equal for 1-target trial data and 2-target trial data, the average ‘correction factor’ (σ2/σ1) from equation 2 is 1.04 when rounded to the nearest hundredths place, and thus has little effect on the main population-averaged results presented in Figure 4b of the manuscript. Please see Author response image 1 in the point-by-point response to Reviewer 1 to see the exact size of the effect for the data from both Expt 2a and Expt 2b. It’s worth noting here as an aside that the average σ2/σ1 ratio quoted above, 1.04, is slightly different from the ratio between the average values of σ2 and σ1, which is 1.00, because in general for paired measurements, the mean of their ratio is different from the ratio of their means, with the former biased upward (biased only mildly if σ1 << µ1 but more dramatically as σ1 grows larger compared to µ1).

**Author response image 1. sa2fig1:** 

Overall, the presentation of this section needs to be made much more accessible – or some analyses, if not absolutely crucial for the main conclusions to hold – should be removed.

We have now revised this section, making it more readable, but at the same time providing additional information that address many of the questions that reviewers 1 and especially 2 raised in their specific comments. Note that key excerpts from these revisions are quoted in section 2.2 of this letter, and that we have provided a redlined version of the revised paper showing key revisions in detail.

The revision should also include a point-to-point response to the concerns raised by the individual reviewers as shown below.

Please see below. We feel that these detailed responses completely address the reviewers’ concerns.

Reviewer #2 (Recommendations for the authors):1. Line 230: during this epoch ◊ which epoch?

We mean to say “during the test epoch”. We have now updated the text with this clarification.

2. Line 288: it is unclear what is being compared here. What is the contrast?

The comparison being made is the prediction index (PI) calculated using the MA prediction and 2-target data vs the π calculated using the PO prediction and 2-target trial data. This comparison is made for samples collected from movement onset until TON and TRESP. We have updated the text to make this clearer, thanks!

3. Figure 5: it should be made clearer that top row is linked to exp2a and bottom one to 2b. Unclear that the labels on the left are for the three panels on that line.

We have made this clearer in the revised version of the figure, thanks!

4. Figure 5c and 5f: the authors might want to use robust correlations to get rid of the influence of outliers rather that doing separate analyses with and without outliers.

To address the reviewer’s concern from Section 2.2, we have completely overhauled Figure 5 and the associated text, to better motivate, present, and formulate the analysis of inter-individual differences that was not as clear as it could have been in the original version of Figure 5.

5. Line 456: the authors should avoid using the word "predict" as they are only using correlation and not out-of-sample predictions.

There must be some confusion here. Line 456 is actually a section heading, which says:

“Performance-optimization predicts motor planning for obstacle avoidance”

However, there are no correlation-based analyses in this section – either in the original version of the manuscript or in the revised version.

That said, we use the word “predicts” in the section heading here because this section in the Results presents the findings illustrated in Figure 4, where we compare experimental data to parameter-free versions of the MA and PO models (equations 1-2), and we find that PO model prediction consistently explains the data from Expts 2a and 2b, whereas the MA model prediction does not. But, as noted, neither model has free parameters, and since cross-validation is used to address the consequences of parameter overfitting, we don’t see how it could be applicable.

As a separate point, it should be noted that the ‘refined’ versions of the MA and PO models (defined in equations 3 and 4), which include one free weighting parameter each, were previously included in the subsequent section, and therefore were not relevant to this comment at the time it was made. But we have, in the revised version of this paper, moved the analysis of these refined models into the “Performance optimization predicts motor planning for obstacle avoidance” section. Note that these single-parameter models are indeed analyzed with out-of-sample predictions (both in the current and previous versions of the manuscript). Also note that moving the refined model analysis to this section was based on the advice of this reviewer, who suggested we present the refined model analysis like the results in Figure 4b (see section 2.4.4 above), and we accomplished this by directly adding those results to 4b.

6. Line 461-468: the motivation behind this analysis is unclear to me. In addition, the concept of polarity seems to be ill-defined (dichotomization of a continuous variable). What are these tests comparing ? How are these comparisons linked to the MA or PO predictions.

We apologize for the confusion here and agree that this could have been better explained, and have made revisions to this section in our revised manuscript to clarify. To be clear, this test is simply comparing the obstacle-free vs obstacle-present 2-target trials IMDs. For the obstacle-present data, left-side and right-side obstacle conditions were combined by aligning the results wrt the obstacle location. This amounts to flipping the left-side data and then averaging it with the right-side data. Thus, to achieve a balanced comparison with the obstacle-free 2-target trial data, for which left-side and rightside obstacle conditions did not exist, we randomly assigned left-side vs right-side obstacle-present labels to our obstacle-free 2-target trial data, and combined in the same manner we did the obstacle-present data. The resulting population-averaged obstacle-free 2-target trial movement direction data are shown in Figure 4b.

7. Line 516-518: robust correlations should be preferred and the authors should display these correlations if they want to make any conclusion on this basis.

As noted in Section 2.10 above, we’ve completely overhauled this section and its associated analyses, and in the revised analysis we’re no longer computing correlation coefficients.

8. Line 520-522: I don't understand how this conclusion follows from the correlation analysis above.

As noted in Section 2.10 above, we’ve completely overhauled this section and its associated analyses.

9. Line 523-530: Mu2i, MuHat2i, MuBar1A are used without any explanations of the Hat, Bar, presence or absence of "i" as index, etc.

We agree that the symbols here could use explanation, and have fixed this in the revision.

10. Line 551-553: this is not a prediction, it is a fit….

The reviewer is correct that the analysis presented here was based on a fit, however the predictive ability of the PO model that we were specifically referring to was based on the correlation analysis discussed in the first paragraph of this section. Nonetheless, this section has been entirely revised, and we hope our changes have made the delineation between predictions and fits clearer.

Reviewer #3 (Recommendations for the authors):The correlation results w/r/t variability (Figure 5) are an impressive extension of the concept of optimal planning and a rather nice discovery. This finding could perhaps be related, in the discussion or introduction, to bayes optimal incorporation of variability in motor planning under risk (e.g. Trommershäuser et al., 2003).

We thank the reviewer for bringing this article to our attention. To highlight previous work demonstrating the incorporation of variability into movement planning, we’ve now cited this paper, along with a few other related articles, in the Discussion section.

A visual schematic of the PO model could be helpful?

We thank the reviewer for the suggestion. We considered this, but frankly we couldn’t think up a schematic design that we thought would be helpful enough to justify its inclusion. We welcome any suggestions.